# CASh: Causality Alignment Shifting to Unveil Vulnerabilities in Visual-Language Model

## Abstract

Existing adversarial attacks on vision-language models (VLMs) primarily use joint occurrence likelihoods to capture interdependency, often missing the true relationship between the text and the image.This paper presents a novel attack, CASh, on VLMs by manipulating latent causal representations between images and text in pre-trained models. We leverage the cross-attention matrix to capture causality alignment and exploit its singular properties to develop an efficient perturbation algorithm that modifies VLM tasks. Our attack targets the core causal relationships that exist independently of specific VLMs, ensuring transferability across models. Unlike existing attacks that primarily perturb inputs using correlation-based patterns, our approach accounts for causality, offering interpretability by showing how causal shifts lead to changes in VLM behavior. We evaluate CASh across various VLMs and compare it to existing attack methods. Our results demonstrate a significant performance boost, with an average improvement of 20.88% in transferable attack capability.

## 1 Introduction

Using the combined power of vision and language, Vision-Language Models (VLMs) have shown effectiveness in capturing the complex interplay between images and text. Thus, they have gained significant popularity in advanced tasks such as vision-language retrieval (VLR) (Cao et al., 2022), visual entailment (VE) (Li et al., 2023), visual grounding (VG) (Hong et al., 2022), Visual Reasoning(VR) (Chen et al., 2023b) and visual question answering (VQA) (Alayrac et al., 2022; Tsimpoukelli et al., 2021).

However, the crucial interdependencies between visual and textual modalities that underpin VLM capabilities also expose a significant vulnerability: these models are highly susceptible to adversarial attacks specifically designed to disrupt inter-modality interactions. Existing attacks can be broadly categorized into two paradigms. The first and more prevalent paradigm, exemplified by works such as SGA (Lu et al., 2023), Co-Attack (Zhang et al., 2022a), VLATTACK (Yin et al., 2024a), and TMM (Wang et al., 2024a), employs surrogate models to craft broadly applicable adversarial examples by exploiting statistical biases like object co-occurrence probabilities in a shared latent space TMM (Wang et al., 2024a; Yin et al., 2024b), This strategy aims to address the challenges of real-world deployment and offers a robust means to test VLM resilience (Lu et al., 2023; Wang et al., 2024a). In contrast, a distinct second paradigm, including studies like (Ying et al., 2025) and (Qi et al., 2024), operates by injecting adversarial perturbations directly into image pixels to jailbreak the model's safety alignment and elicit harmful responses, often without explicitly targeting nuanced inter-modal relationships.

Despite the effectiveness of the predominant co-occurrence-based approach, it suffers from two fundamental limitations. First, the exploited co-occurrence probabilities often reflect superficial statistical regularities rather than meaningful semantic relationships. For instance, in a household setting, images of both a 'dog' and a 'cat' might frequently appear indoors, potentially leading a model to erroneously associate a 'dog' image with the text 'a cat on the sofa' based solely on this environmental correlation rather than genuine visual evidence. Second, and more critically, this reliance on shallow, dataset-specific correlations severely hinders attack transferability. Generating effective transferable attacks requires capturing robust, invariant inter-modal dependencies across diverse tasks and modalities. However, current co-occurrence-based methods primarily capture

transient patterns from their training data, which fail to generalize to unseen data distributions or related tasks, resulting in weak transferability across different VLM architectures.

To address these limits, we closely explored the inference mechanisms of the open-source VLMs (Li et al., 2021; Yang et al., 2022; Wu et al., 2024b; Kim et al., 2021; Bai et al., 2025; Xiao et al., 2023) and observed that the core challenge in designing effective attacks against VLMs is achieving fine-grained relationship alignment. An adversary must identify exactly which image regions correspond to specific textual cues, then inject perturbations that subvert the model's joint reasoning without introducing conspicuous artifacts. Such precise alignment is crucial because it ensures that perturbations target the intrinsic semantic structure actually used by the model. This allows high attack success with minimal changes while preserving stealthiness, and also improves transferability, since these causal relationships are more likely to remain invariant across different VLM architectures than shallow correlations. Motivated by this observation, we adopt a causal perspective using Pearl's Structural Causal Model (SCM) (Pearl, 2009) and recent extensions (Wang et al., 2025). This approach lets us explicitly represent the causal links between image regions and textual tokens. It further allows us to distinguish true causal effects from spurious patterns and to formulate targeted counterfactual interventions. Earlier studies (Peng & Wei, 2024; Abbasnejad et al., 2020; Schölkopf et al., 2021; Yang et al., 2021; Wang et al., 2024b) build structural causal models or use counterfactual interventions to examine how certain words direct attention to specific visual regions, or how visual cues shape language output. Our work takes a different perspective by focusing on causality at the token or object level. Here, the nodes in the causal graph correspond to concrete visual or linguistic elements, not high-level abstract variables. This is different from traditional causal modeling in fields like medicine or social science, where interventions are defined at a macro level and directly related to overall outcomes.

To this end, we propose **CASh** (Causality Alignment Shifting Attack), a novel attack method that perturbs latent causal representations in pre-trained vision-language models. Our approach models causal dependencies within and across modalities using SCMs, and aligns image and text features via a regularized cross-attention mechanism. We then identify the high-impact directions in the causal alignment space using singular value decomposition (SVD) and inject norm-bounded perturbations to selectively break these links. Although the perturbations minimally modify the input, they severely misalign the model's causal connections, resulting in marked deterioration across diverse downstream tasks.

For example, Figure 1 shows that "*kitchen*" and "*microwave*" are connected via the mediating concept "*food*". When we perturb the image and replace "*kitchen*" with "*room*" in the text using CASh, the model's cross-modal alignment breaks down, leading it to change its answer from "*Yes*" to "*No*".

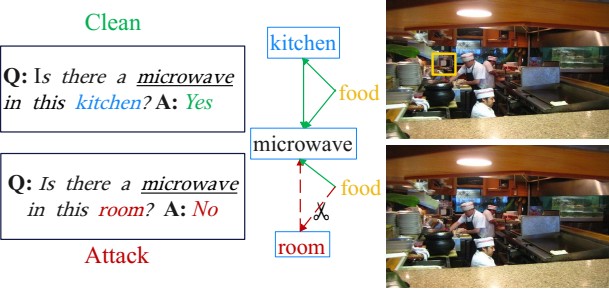

Figure 1: Causal Relationship Alignment Shift.

In summary, our primary contributions include: 1) **First Attempt to Exploit Causality for VLM Adversarial Attacks:** To the best of our knowledge, this is the first work that enhances adversarial attacks by explicitly shifting the causality alignment between image and text, rather than merely perturbing their raw statistical features. 2) **Causal Dependency Quantification:** We propose the novel application of cross-attention matrices as a diagnostic tool for quantifying causal dependencies across multimodel representations. 3) **Good Experimental Performance:** We evaluate the effectiveness of the CASh method across multiple downstream tasks, revealing significant performance degradation. Our findings highlight the need for improved causality alignment mechanisms to strengthen the robustness of VLMs against such threats.

## 2 RELATED WORK

### 2.1 CAUSAL RELATIONSHIPS IN VLMS

Understanding causal relationships in VLMs has gained attention as a means to move beyond correlation-driven predictions and improve robustness in multimodal tasks. Foundational works on causality (Pearl, 2009; Schölkopf et al., 2012) have inspired studies like (Lopez-Paz et al., 2015),

which delve into counterfactual reasoning to distinguish genuine cause-effect relationships from spurious associations in multimodal datasets. Additionally, (Yang et al., 2021) introduced causal attention mechanisms, significantly improving performance in tasks such as image captioning by modeling the cause-effect dynamics between visual regions and textual tokens. Likewise, (Wang et al., 2024b) developed a causal learning framework for vision-and-language navigation, using SCMs to disentangle spurious correlations in navigation instructions and visual scenes, achieving better generalization across diverse environments. Recent benchmarks, including Causal3D (Liu et al., 2025), further highlight the importance of causality in VLMs by providing datasets with structured causal graphs to evaluate models on complex visual reasoning tasks. Furthermore, for visual question answering tasks, (Chen et al., 2024) proposed a causal intervention framework enabling models to make predictions based on causal relationships rather than spurious correlations.

## 2.2 Adversarial Attacks in VLMs

Adversarial attacks on VLMs have exposed significant vulnerabilities, particularly in their multimodal nature, where attackers exploit the visual modality to bypass safety mechanisms. Previous research has utilized gradient-based methods to generate perturbations across image and text modalities, assuming full model accessibility, as demonstrated in works such as (Luo et al., 2024; Gao et al., 2024; Wu et al., 2024a). Other studies (Wang et al., 2024a; Zhao et al., 2023b; Dong et al., 2023; Wang et al., 2024c; Wu et al., 2024a) have explored scenarios where attackers with limited VLM knowledge employ surrogate models to target other systems, with (Wang et al., 2024a) employing attention-directed feature perturbation and (Zhao et al., 2023b) leveraging pretrained CLIP and BLIP models. Additionally, (Chen et al., 2023a) further explored adaptive ensemble attacks, showing that synchronizing outputs from diverse surrogate models, can amplify transferability across architectures like CNNs and ViTs. Similarly, (Chen et al., 2025) proposed a multimodal feature heterogeneous attack framework, leveraging triplet contrastive learning to enhance the transferability of adversarial examples across medical imaging VLMs, highlighting the underutilization of modal differences in prior attacks. Moreover, (Zhao et al., 2023a) evaluated the robustness of large VLMs under black-box settings, crafting targeted adversarial examples against models like CLIP and transferring them to others like LLaVA, revealing the ease of deceiving VLMs into producing incorrect outputs. However, these methods focus on shallow features and overlook deeper model structures. In contrast, our transfer-based attack leverages the causal relationships between text and images to enhance effectiveness and transferability.

## 2.3 Causality-Informed Adversarial Attacks

Prior causality-driven adversarial attacks fall into several conceptual categories. Methods such as CausalAdv (Zhang et al., 2022b) pursue distribution-level causal alignment, aiming to reduce discrepancies between natural and adversarial samples to improve robustness. Others, including the CADE framework (Cai et al., 2024), employ counterfactual reasoning, treating adversarial examples as alternative causal worlds and optimizing perturbations through explicit manipulation of causal variables. In addition, the paper (Koyuncu et al., 2023) studies adversarial vulnerabilities in causal inference systems, focusing on how attacks distort causal estimators rather than multimodal representations. While these approaches leverage causality to preserve or model causal structure, our work takes a complementary perspective. We intentionally exploit causal misalignment in multimodal cross-attention, targeting the mechanism that binds visual and textual entities. This reframing enables more effective disruption of semantic grounding and yields adversarial examples with stronger transferability. Our method thus extends the space of causality-informed attacks by treating causal dependencies as exploitable weaknesses rather than constraints to maintain.

## 3 The CASh Attack

In this section, we specifically introduce our CASh attack method. Let $\mathcal{F}^s$ be a publicly available, pre-trained VLM, and $\mathcal{F}^a$ denote the unknown, black-box target VLM. Given a clean image–text pair $(I, T)$ and norm bounds $\epsilon_I, \epsilon_T$, we solve

$$(I', T') = \mathcal{F}^s(A^G, \mathcal{G}^I(I, \delta_I), \mathcal{G}^T(T, \delta_T)),$$
$$\|\delta_I\|_\infty \leq \epsilon_I, \ \|\delta_T\|_\infty \leq \epsilon_T, \tag{1}$$

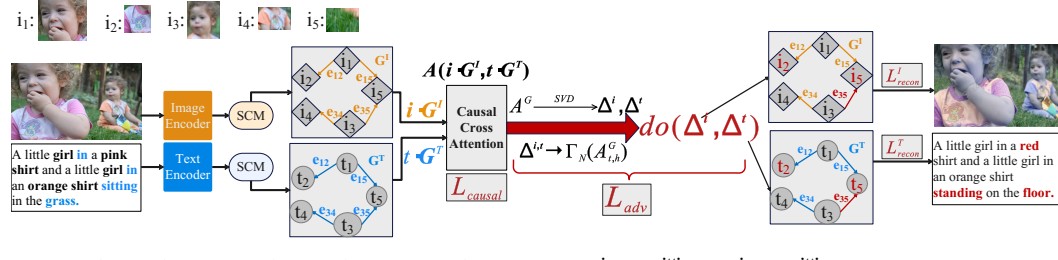

Figure 2: CASh Framework. Inputs are encoded via Image and Text Encoders, processed with SCMs, and aligned using Cross Attention ($A(\mathbf{i} \cdot \mathbf{G}^I, \mathbf{t} \cdot \mathbf{G}^T)$). Causal Attention ($A^G$) and SVD generate perturbations (($\mathbf{\Delta}^i, \mathbf{\Delta}^t$)) with the do-operator, guided by $L_{\text{causal}}$ and $L_{\text{adv}}$ losses, while $L_{\text{recon}}$ produces a manipulated output (e.g., "A little girl in a red shirt... standing on the floor").

where $G^I, G^T, A^G$ is our attack strategy chosen so that $\mathcal{F}^s(I, T)$ produces some $(I', T')$. By transferability, the resulting adversarial example $(I', T')$ satisfies $\mathcal{F}^a(I', T') \notin \mathcal{Y}$, even though the attacker has no access to the architectures, parameters, or training data of the $\mathcal{F}^a$.

Our CASh attack procedure is shown in Fig. 2. The goal of the attacker is to shift the alignment relationship from inherent causality in the image and text modalities. Specifically, we integrate an image $I$ and its text description $T$ (e.g., "*a little girl in a pink shirt and a little girl in an orange shirt sitting on the grass.*"). The encoded features $\mathbf{i}$ and $\mathbf{t}$ are extracted through $\mathcal{F}^s$'s image or text encoding blocks. We then apply structural causal relationships ($\mathbf{G}^I, \mathbf{G}^T$) captured by our SCM (Section 3.1) to these features, which are aligned via a causal cross-attention module computing $A^G = A(\mathbf{i} \cdot \mathbf{G}^I, \mathbf{t} \cdot \mathbf{G}^T)$, loss $\mathcal{L}_{\text{causal}}$. The causal loss $\mathcal{L}_{\text{causal}}$ ensures robust causality capture in cross-attention (Section 3.2). Then we find the most possible attack direction by SVD decomposition of the causal cross-attention matrix, with the perturbation lower bound guaranteed by Theorem 1. Next, an adversarial intervention $do(\Delta^\mathbf{i}, \Delta^\mathbf{t})$ perturbs modality latent features of both modalities. Finally, the adversarial samples $I'$ and $T'$ are reconstructed by jointly optimizing losses $\mathcal{L}^I_{\text{recon}}$ and $\mathcal{L}^T_{\text{recon}}$ (see Section 3.3). For instance, the attacker changes $\mathbf{e}_{35}$ from "*sitting*" to "*standing*" or changes $\mathbf{t}_5$ from the "*grass*" to "*floor*". As a result, the causal relationship between the text semantic information and the image space is misaligned, leading to the generated text not matching the image.

## 3.1 Modeling Causal Relationships in Image and Text

Current attack methods (Lu et al., 2023; Zhang et al., 2022a; Yin et al., 2024a), which depend entirely on VLM backbones (e.g. ViT (Dosovitskiy et al., 2021) and CNN (LeCun et al., 1989)), can only exploit superficial co-occurrence patterns, missing specific critical causal structures such as "*sunlight angle*" → "*shadow formation*" or "*rain*" leads to "*wet ground*", thus leading to reduced attack efficiency in disrupting critical cause-effect dynamics. However, images $I$ and texts $T$ are compositional, with entities (e.g., "*car*") and relations (e.g., "*car on road*") implying causal structures that current VLM encoders, such as CNNs, RNNs, or Transformers, collapse into flat vectors, hindering correlation-based attacks. In contrast, SCMs model modalities as DAGs (Thulasiraman & Swamy, 1992) whose nodes and edges encode entities and causal dependencies, exposing vulnerabilities that enable more effective attacks than correlation-only approaches.

In this way, we construct the causal graphs for the image and caption pair $(I, T), I = \{I_i\}_{i=1}^{N_I}, T = \{T_i\}_{T=1}^{N_T}$. The text causal graph $\mathcal{G}^T$ is built first in its entirety, followed by the image causal graph $\mathcal{G}^I$, which uses only the already-extracted textual entities as open-vocabulary anchors for visual grounding.

### 3.1.1 Construct Text Causal Graph $\mathcal{G}^T$

Given an image caption $T$, we extract full noun phrases as primary entities, including determiners, adjectives, compound nouns, and prepositional modifiers (e.g., "a large black cat on the red sofa", "bright sunlight through the window"). Named entities recognized by Stanza (Qi et al., 2020) are also included as standalone nodes even when overlapping with noun phrases. Each entity $T_j$ is represented in two forms: (i) **Surface span** (full phrase): used as the open-vocabulary query for

visual grounding with Grounding DINO (Liu et al., 2023) and for feature extraction $\mathbf{t}_j = \text{BERT}(T_j)$ of node $T_j$ using a frozen BERT-large-uncased model (Devlin et al., 2019); (ii) **Head lemma** (e.g., "cat", "sunlight", "sofa"): assigned as canonical label $l_j$ obtained by Stanza's built-in head detection within the phrase, used to instantiate physical causal edges. This dual representation enables highly accurate cross-modal grounding via rich descriptive phrases and reliable physical causality lookup using clean head nouns.

The adjacency matrix of the text modality $\mathbf{G}^T \in \{0, 1\}^{N_T \times N_T}$ is obtained by taking the **union** of two complementary sources of causal relations:

$$\mathbf{G}^T = \mathbf{G}^{T,\text{syn}} \vee \mathbf{G}^{T,\text{phys}} \quad \equiv \quad \mathbf{G}_{kj}^T = \begin{cases} 1, & \text{if } \mathbf{G}_{kj}^{T,\text{syn}} = 1 \text{ or } \mathbf{G}_{kj}^{T,\text{phys}} = 1, \\ 0, & \text{otherwise.} \end{cases} \tag{2}$$

$\mathbf{G}^{T,\text{syn}}$ encodes linguistic causality derived directly from Stanza's dependency parsing (Qi et al., 2020). A directed edge $k \to j$ ($\mathbf{G}_{kj}^{T,\text{syn}} = 1$) is placed whenever node $\mathbf{t}_k$ is the syntactic head of node $\mathbf{t}_j$ in the dependency tree. $\mathbf{G}_{\text{phys}}^T$ incorporates real-world physical commonsense via the PhysCause-1037 lexicon (Zhang et al., 2025). For each ordered pair of text entities $(T_k, T_j)$ with canonical head lemmas $(l_k, l_j)$, we query the lexicon: if the rule $l_k \to l_j$ exists (e.g., "sun $\to$ sunlight", "floor $\to$ cat"), the corresponding edge is added ($\mathbf{G}_{kj}^{T,\text{phys}} = 1$). Empirically, syntactic and physical edges almost never conflict ($< 0.8\%$ cases on standard datasets). Taking their union thus yields a richer causal graph that seamlessly integrates linguistic structure and physical commonsense, with no observable contradictions. Then, the $\mathcal{G}^T$ is represented as:

$$\mathcal{G}_j^T\big(\text{Pa}(T_j), \eta_j\big) = \sum_{k \in \text{Pa}(T_j)} \mathbf{G}_{jk}^T \mathbf{t}_k + \eta_j, \tag{3}$$

where $\text{Pa}(T_j)$ denotes the parent nodes and $\eta_j \sim \mathcal{N}(0, \sigma_T^2)$, which is node-specific exogenous noise capturing unmodeled factors.

### 3.1.2 CONSTRUCT TEXT CAUSAL GRAPH $\mathcal{G}^I$

Given the image $I$, we use Grounding DINO to extract image entities nodes $I_i$ according to the set of canonical text labels $\{l_j\}_{j=1}^{N_T}$ obtained from Head lemma during constructing text causal graph. The feature $\mathbf{i}_j$ of each image entities nodes $I_j$ is encoded by frozen CLIP ViT-L/14 (Radford et al., 2021), $\mathbf{i}_i = \text{ViT}(I_i)$. The adjacency matrix of the text modality $\mathbf{G}^I \in \{0, 1\}^{N_I \times N_I}$ is same as $\mathbf{G}^T$ due to they are benefit image-text pair. Therefore, the $\mathcal{G}^I$ is represented as:

$$\mathcal{G}_i^I\big(\text{Pa}(I_i), \gamma_i\big) = \sum_{j \in \text{Pa}(I_i)} \mathbf{G}_{ij}^I \mathbf{i}_j + \gamma_i, \tag{4}$$

where $\text{Pa}(I_i)$ denotes parent nodes and $\gamma_i \sim \mathcal{N}(0, \sigma_I^2)$. This ensures (i) complete independence of the linguistic causal graph, (ii) perfect semantic alignment between modalities via shared labels, and (iii) maximal interpretability by separating linguistic and physical causal mechanisms exactly where they belong.

### 3.2 CROSS-ATTENTION FOR ALIGNMENT

A key challenge in VLMs is aligning diverse image and text modalities, where cross-attention assigns fine-grained weights $\alpha_{ij}$ linking nodes (e.g., the textual "*sun*" to its image region), enabling highly targeted attacks with minimal perturbations. Compared to coarse global alignment, embedding causal relationships from SCMs via regularization (see Eq. (5)) further boosts attack success and interpretability, while simultaneously exposing vulnerabilities that attackers can exploit. Accordingly, we employ causal cross-attention to align the $\mathcal{G}_I$ and $\mathcal{G}_T$ based on their causal adjacency matrices $\mathbf{G}^I$ and $\mathbf{G}^T$ by $\alpha_{ij} = \text{softmax}_j(A_{ij}^G)$, $A_{ij}^G = \frac{(W_Q(\mathbf{i}_i \cdot \mathbf{G}^I))^\top (W_K(\mathbf{t}_j \cdot \mathbf{G}^T))}{\sqrt{D'}}$, where attention score matrix $A^G \in \mathbb{R}^{N_I \times N_T}$ captures the raw similarity between nodes, $\alpha_{ij}$ provides the weighted distribution for integration, $W_Q, W_K \in \mathbb{R}^{D' \times D}$ projects the features into a shared space (with $D'$ as the projected dimension), while $\text{softmax}_j$ normalizes over $j$ to ensure $\sum_j \alpha_{ij} = 1$, and the attended representation is $\mathbf{i}_i' = \sum_j \alpha_{ij}(W_V(\mathbf{t}_j \cdot \mathbf{G}^T))$. To ensure generalization and adaptability across different downstream

tasks, we define $\mathbf{M}_{1,2}$ as the causal features of either image or text modalities ($\mathbf{i} \cdot \mathbf{G}^I$ or $\mathbf{t} \cdot \mathbf{G}^T$). Thus, the causal cross-attention can be expressed as $A_{ij}^G = \frac{(W_Q\mathbf{M}_1)^\top (W_K\mathbf{M}_2)}{\sqrt{D'}}$. This alignment relationship provides an attacker with a direct target, as perturbing attention weights can effectively disrupt causal alignments between $\mathbf{M}_1$ and $\mathbf{M}_2$ from two different modalities, thereby exposing VLM vulnerabilities. To ensure the alignment adheres to the causal structure, we introduce a **causal regularization term**:

$$\mathcal{L}_{\text{causal}} = \sum_{i,k} \|\alpha_{ik} - \sum_{j \in \text{Pa}(k)} \alpha_{ij} \mathbf{G}_{\mathbf{M}_2}(j,k)\|_2^2, \tag{5}$$

where $\alpha_{i\cdot}$ is the $i$-th causal feature from $\mathbf{M}_1$. This formulation ensures that alignments strictly adhere to causal pathways, as demonstrated through three fundamental reasoning patterns: 1)Deductive Reasoning: Explicit rule-based alignment(e.g., mapping visual *"shadow"* to textual *"sunlight"* through the causal rule *"if sun, then shadow"* ). 2)Inductive Reasoning: Pattern generalization across instances(e.g., consistent alignment of *"cars on roads"* relationships). 3)Abductive Reasoning: Explanatory inference from observations (e.g., deducing a light source from shadow features and the textual *"sunlight"* concept).

### 3.3 EFFECTIVE CROSS-ATTENTION ATTACK

For effectively adding perturbation, we consider two points: (1) which target layers yield perturbations that both fool the surrogate model and transfer well to black-box VLMs, and (2) how to find perturbations that trigger attacks more efficiently. We tackle these by (i) **exploring the core attack direction** of the causality relationship matrix, which captures the coherent text-image relationship, thereby enhancing transferability and (ii) **exploring perturbation bound** to ensure more efficient discovery of adversarial-triggering perturbations.

#### 3.3.1 EXPLORING THE CORE ATTACK DIRECTION

A simple method to disrupt causality in cross-attention mechanisms is to maximize the difference between the adversarial attention matrix $A^{G'}$ and the original $A^G$, typically using a loss function like $\mathcal{L}_{\text{naïve}} = -\|A^{G'} - A^G\|_F^2$. However, this naïve approach lacks effectiveness, efficiency, and stability because it does not target the most influential components of attention alignment and cannot discover perturbations that trigger faster adversarial attacks after locating the most likely attack component. For instance, some changes may be large in magnitude but low in functional impact, meaning that the model might still preserve its decision-making capabilities despite a large perturbation in $A^G$. Crucially, it ignores attention matrix structure, often perturbing low-impact directions. An optimal attack must instead target the most sensitive alignment directions to ensure both substantial magnitude and maximal reasoning disruption.

To address the issues, we introduce a more structured attack strategy that leverages SVD decomposition of the attention matrix $A^G = U\Sigma V^\top$, where $\Sigma = \text{diag}(\lambda_1, \lambda_2, \ldots, \lambda_n)$, $U \in \mathbb{R}^{N_{\mathbf{M}_1} \times r}$, $V \in \mathbb{R}^{N_{\mathbf{M}_2} \times r}$, and $r = \min(N_{\mathbf{M}_1}, N_{\mathbf{M}_2})$. We select the top-$k$ singular components $A_k^G = U_k \Sigma_k V_k^\top$ to identify critical alignment relationships, then project them to generate perturbations $\Delta_i^{\mathbf{M}_1}$ and $\Delta_j^{\mathbf{M}_2}$ for image and text features by

$$\Delta_i^{\mathbf{M}_1} = \sum_{n=1}^{k} \lambda_n (\mathbf{M}_1^k)_{in} (W_Q^\top \mathbf{u}_n), \quad \Delta_j^{\mathbf{M}_2} = \sum_{n=1}^{k} \lambda_n (\mathbf{M}_2^k)_{jn} (W_K^\top \mathbf{v}_n), \tag{6}$$

where $\mathbf{u}_n$ and $\mathbf{v}_n$ are basis vectors, constraints $\sum_i \|\Delta_i^{\mathbf{M}_1}\|_2^2 \leq \epsilon_{\mathbf{M}_1}$, $\sum_j \|\Delta_j^{\mathbf{M}_2}\|_2^2 \leq \epsilon_{\mathbf{M}_2}$ are enforced via projection. With $\Delta_i^{\mathbf{M}_1}$ and $\Delta_j^{\mathbf{M}_2}$ determined in Eq. (6), we map them back to input perturbations $\delta_I$ and $\delta_T$ to generate $I'$ and $T'$, we optimize the **reconstruction loss**:

$$\mathcal{L}_{\text{recon}}^I = \sum_i \|\mathcal{F}^s(I') - (\mathbf{i} + \Delta_i^{\mathbf{M}_I})\|_2^2, \quad \mathcal{L}_{\text{recon}}^T = \sum_j \|\mathcal{F}^s(T') - (\mathbf{t}_j + \Delta_j^{\mathbf{M}_T})\|_2^2. \tag{7}$$

For texts, since $T$ is discrete, we via discrete search (e.g., synonym replacement) to approximate $\delta_T$ get $(T, \delta_T) \rightarrow T'$. This approach ensures targeted causality shifting by leveraging the perturbation

matrix's spectral properties. For input layer, we employ BERT-Attack (Li et al., 2020) to generate adversarial text $T'$ as input text and perturbated image calculated by adding perturbation $\delta_I$ to generate $I'$.

### 3.3.2 EXPLORING PERTURBATION BOUND

Instead of applying arbitrarily large perturbations, we quantify perturbation sensitivity and provide a theoretical justification for the attack strategy. In VLM models with $L$ layers and one layer containing $H$ heads, capture different features from the alignment relation in cross-attention, the perturbation introduced at the input layer propagates through to the output layer. As a result, the entire space is often considered during the search process, which significantly increases computational cost. To address this issue, we analyze the perturbation for layers according to the Theorem 1 (See Appendix A.1), the perturbation sensitivity of causality relationship $A_{l,h}^G$ in layer $l$ and head $h$ satisfy

$$\Gamma_{\mathcal{N}}(A_{l,h}^G) \geq \left( \frac{\|W_K\|_F^2}{\|W_K\|_2^2} + \frac{\|W_Q\|_F^2}{\|W_Q\|_2^2} \right)_{l,h}. \tag{8}$$

Therefore, we set the low bound to initial value of perturbation $\Delta_{A_{l,h}^G}$ for the $A_{l,h}^G$, the perturbation sensitivity bound for layer $l$ is calculated by $\Gamma_{\mathcal{N}}(A_l^G) = \sum_{h=1}^H \Gamma_{\mathcal{N}}(A_{l,h}^G), \quad l = \{1, 2, \dots, L\}$. Then we find the most vulnerable target layer $t$ by $A_t^G = \arg\max_l \left( \Gamma_{\mathcal{N}}(A_1^G), \dots, \Gamma_{\mathcal{N}}(A_L^G) \right)$. This identifies the layer $t$ will experience the greatest change in output with perturbations in its input. From Theorem 1 and Eq. (6), we simulate the effect of intervention denoted as $do(\cdot)$ in a causal model and derive the KL-divergence between the original output distribution $P(\mathcal{Y})$ and the post-intervention distribution $P(\mathcal{Y}' \mid do(A_t^G = A_t^{G'}))$ as follows(specific calculation see Appendix A.5),

$$D_{\text{KL}}\left( P(\mathcal{Y}' \mid do(A_t^G = A_t^{G'})) \parallel P(\mathcal{Y}) \right) = D_{\text{KL}}\left( P(\mathcal{Y}' \mid do(\{ \Delta_i^{h,\mathbf{M}_1}, \Delta_j^{h,\mathbf{M}_2} \}_{h=1}^H)) \parallel P(\mathcal{Y}) \right). \tag{9}$$

We maximize the KL divergence for making the perturbed distribution as different as possible from the original distribution, defining this loss as **adversarial effect loss**:

$$\mathcal{L}_{\text{adv}} = -D_{\text{KL}}(P(\mathcal{Y}' \mid do(A^G = A_t^{G'})) \parallel P(\mathcal{Y})). \tag{10}$$

We use $\beta_1$ and $\beta_2$ to control the perturbation strength when optimize overall loss function :

$$\arg \min_{\delta_{I,T}, \Delta_h^{\mathbf{M}_{1,2}}} \mathcal{L}_{\text{adv}} + \mathcal{L}_{\text{causal}} + \beta_1 \mathcal{L}_{\text{recon}}^I + \beta_2 \mathcal{L}_{\text{recon}}^T. \tag{11}$$

$$\text{s.t.} \quad \|\delta_T\|_\infty \leq \epsilon_T, \|\delta_I\|_\infty \leq \epsilon_I, \quad \|\Delta_h^{\mathbf{M}_{1,2}}\|_\infty \geq \Gamma_{\mathcal{N}}(A_{t,h}^G), \ h = 1, 2, \dots, H.$$

## 4 EXPERIMENTS

In this section, we present the experimental results evaluating the effectiveness of our proposed attack strategy on these VLM downstream tasks: VLR, VE, VG, VR and VQA.

### 4.1 EXPERIMENTAL SETTINGS

**Datasets:** Based on the above five downstream tasks, we collect 1000 test samples from Flickr30K and 5000 val samples from MSCOCO for VLR task. For varifying the causality performance of our attack method, we select all test samples from SNLI-VE for VE task and randomly select 5000 test samples from NLVR2 for VR task. Furthermore, we randomly select 5000 val samples from VQAv2 for VQA task. For VG task, we select all TestA and TestB samples from RefCOCO+ to test the performance. For ablation, we use VQA-CP v2 to leverage its strong language priors for verifying our causality-driven attack.

**VLM Models and Attack Methods:** All above dataset will used on several popular vision-language models, including ALBEF (Li et al., 2021), TCL (Yang et al., 2022), DeepSeek-VL2 (DS-VL2) (Wu et al., 2024b), ViLT (Kim et al., 2021), Qwen2.5-VL (QW-VL) (Bai et al., 2025), Florence-2 (Flor2) (Xiao et al., 2023). To validate our attack's practical threat, we evaluate its transferability on leading black-box API models, OpenAI's GPT-4o (GPT) (OpenAI, 2024) and Google's Gemini 2.5-Pro (Gemini) (Comanici et al., 2025), the most advanced commercial VLMs. For demonstrating our attack method, we compare our proposed method CASh with current multi-model attack methods

considering the multimodal features in VLM model, such as SGA (Lu et al., 2023), Co-Attack (Zhang et al., 2022a), VLATTACK (Yin et al., 2024a), TMM (Wang et al., 2024a), CMI-Attack (Fu et al., 2024), VQATTACK (Yin et al., 2024b).

**Evaluation Metrics:** In this work, we utilize the Attack Success Rate (ASR) metric to evaluate both the efficacy of white-box attacks and the transferability of black-box attacks against VLP models. We also use IR and TR to denote the percentage of top-1 image and subtitle retrievals, respectively, that fail to include the correct match. Bold values indicate the best performance in each column throughout all tables.

**Implementation Details:** We use VLMs with cross-attention blocks (e.g. ALBEF, TCL, DS-VL2,QW-VL) as a surrogate model, without finetuning, to attack white- and black-box VLMs. For images, we set the perturbation bound $\epsilon_I = 16/255$ to control adversarial noise, and for text, we apply bounded lexical substitutions $\epsilon_T = 1$ based on semantic similarity. To improve robustness and transferability, we incorporate a momentum term $\mu = 1$ in gradient updates. This setup ensures a fair evaluation of our causality-based attack on cross-modal retrieval. In our experiments, we set $\beta_1 = 0.05$ and $\beta_2 = 0.1$, which achieve strong attack performance. All experiments run on an NVIDIA A100 GPU with 40GB.

Table 1: The ASR(%) results of VLR on Flickr30K datasets.

| Source | Attack | ALBEF | | TCL | | DS-VL2 | | ViLT | | QW-VL | | Flor2 | |
|---|---|---|---|---|---|---|---|---|---|---|---|---|---|
| | | TR | IR | TR | IR | TR | IR | TR | IR | TR | IR | TR | IR |
| ALBEF | Co-Attack | 77.16 | 83.86 | 15.21 | 29.49 | 10.21 | 12.51 | 20.31 | 23.54 | 11.34 | 12.65 | 13.65 | 12.75 |
| | SGA | 97.24 | 97.28 | 45.42 | 55.25 | 43.31 | 53.21 | 42.21 | 54.56 | 41.34 | 53.32 | 43.65 | 54.35 |
| | CMI-Attack | 97.08 | 97.43 | 62.17 | 69.64 | 65.11 | 64.21 | 51.54 | 53.54 | 65.32 | 67.43 | 62.87 | 64.35 |
| | TMM | 97.53 | 97.51 | 64.97 | 69.60 | 62.76 | 65.31 | 64.21 | 63.21 | 67.82 | 64.82 | 72.13 | 73.14 |
| | **CASh(Ours)** | **98.21** | **98.03** | **72.29** | **76.01** | **65.68** | **66.21** | **68.68** | **64.23** | **72.43** | **79.68** | **82.43** | **79.31** |
| TCL | Co-Attack | 23.15 | 40.04 | 77.94 | 85.59 | 22.34 | 43.25 | 32.14 | 23.15 | 31.24 | 33.21 | 41.52 | 43.21 |
| | SGA | 48.91 | 60.34 | 98.37 | 98.81 | 49.12 | 58.19 | 50.31 | 60.21 | 58.12 | 57.13 | 59.12 | 59.34 |
| | CMI-Attack | 61.52 | 71.73 | 98.00 | 98.67 | 62.31 | 72.31 | 62.34 | 72.31 | 61.34 | 69.34 | 54.31 | 58.42 |
| | TMM | 68.10 | 72.30 | 97.87 | 97.60 | 69.12 | 73.14 | 72.34 | 71.45 | 72.45 | 69.32 | 65.31 | 66.31 |
| | **CASh(Ours)** | **72.43** | **76.32** | **99.12** | **98.89** | **70.08** | **73.18** | **80.34** | **81.23** | **83.15** | **84.35** | **82.11** | **79.13** |
| DS-VL2 | Co-Attack | 21.35 | 34.62 | 22.45 | 35.62 | 78.62 | 84.35 | 23.45 | 24.65 | 22.98 | 31.24 | 26.39 | 30.13 |
| | SGA | 43.21 | 55.34 | 56.32 | 57.82 | 98.32 | 98.45 | 47.89 | 52.37 | 43.56 | 55.12 | 49.75 | 58.24 |
| | CMI-Attack | 62.34 | 63.42 | 71.23 | 70.34 | 98.56 | 98.02 | 64.77 | 59.45 | 61.08 | 67.93 | 56.21 | 63.68 |
| | TMM | 69.32 | 72.34 | 73.41 | 72.54 | 98.01 | 98.32 | 71.04 | 72.50 | 64.86 | 68.62 | 66.19 | 70.33 |
| | **CASh(Ours)** | **73.45** | **75.62** | **85.62** | **83.42** | **98.98** | **99.01** | **80.35** | **82.34** | **80.52** | **83.24** | **78.32** | **79.34** |
| QW-VL | Co-Attack | 22.25 | 33.52 | 32.65 | 38.62 | 25.61 | 31.45 | 34.25 | 32.61 | 75.36 | 83.21 | 35.62 | 32.14 |
| | SGA | 48.75 | 56.23 | 52.41 | 59.06 | 58.29 | 59.13 | 60.21 | 60.32 | 98.35 | 98.76 | 62.35 | 63.12 |
| | CMI-Attack | 62.35 | 65.18 | 68.94 | 61.07 | 66.52 | 69.83 | 63.40 | 67.29 | 97.32 | 97.31 | 72.14 | 73.14 |
| | TMM | 72.56 | 75.39 | 78.14 | 71.82 | 76.03 | 79.47 | 73.25 | 77.68 | 98.99 | 97.97 | 80.12 | 79.34 |
| | **CASh(Ours)** | **80.19** | **89.32** | **90.31** | **90.45** | **89.23** | **88.34** | **85.23** | **87.34** | **99.12** | **99.32** | **85.36** | **89.13** |

Table 2: The ASR (%) results of VG tasks on RefCOCO+ datasets.

| Attack | ALBEF | | TCL | | DS-VL2 | | QW-VL | | Flor2 | |
|---|---|---|---|---|---|---|---|---|---|---|
| | TestA | TestB | TestA | TestB | TestA | TestB | TestA | TestB | TestA | TestB |
| Co-Attack | 22.33 | 13.99 | 40.52 | 34.18 | 32.45 | 34.52 | 40.23 | 35.67 | 34.21 | 30.14 |
| VLATTACK | 25.36 | 24.35 | 41.24 | 35.67 | 32.54 | 35.67 | 36.87 | 36.87 | 36.72 | 30.12 |
| SGA | 55.74 | 50.63 | 43.53 | 37.74 | 45.67 | 51.24 | 38.62 | 40.13 | 42.51 | 41.23 |
| TMM | 67.14 | 59.26 | 57.49 | 50.85 | 52.34 | 51.34 | 53.45 | 53.47 | 62.31 | 61.15 |
| **CASh(Ours)** | **69.32** | **60.35** | **62.34** | **61.43** | **68.35** | **65.41** | **60.32** | **58.92** | **64.31** | **65.32** |

## 4.2 ATTACKING PERFORMANCE

This section details extensive experimental evaluations to demonstrate our attack performance via VLR, VG, VE, VQA, and VR tasks. The performance of the models and the different attack methods is shown in the Tables, our method consistently outperforms the other attack strategies across all models and datasets for both TR and IR tasks.

**1) Significant Advantage in White-box Attacks**: In Table 1, gray shading shows our attack method performance under white-box attacks, our method achieves the highest ASR in both TR and IR tasks.

Table 3: ASR (%) for VQA, VE and VR on VQAv2, TextVQA, SNLI-VE and NLVR2 datasets.

| Tasks | Dataset | Attack | ALBEF | TCL | DS-VL2 | ViLT | QW-VL | Flor2 | GPT | Gemini |
|---|---|---|---|---|---|---|---|---|---|---|
| VQA | VQAv2 | Co-Attack | 11.36 | 27.24 | 15.61 | 19.34 | 22.35 | 24.56 | 10.15 | 10.56 |
| | | VQAttack | 21.60 | 61.32 | 25.67 | 25.67 | 37.68 | 32.61 | 11.25 | 10.98 |
| | | VLATTACK | 48.35 | 55.32 | 48.93 | 49.54 | 50.13 | 49.87 | 15.21 | 13.91 |
| | | **CASh(Ours)** | **69.23** | **62.12** | **65.78** | **60.98** | **67.89** | **62.89** | **22.89** | **23.29** |
| | TextVQA | Co-Attack | 23.45 | 20.50 | 40.68 | 38.56 | 39.32 | 40.35 | 20.34 | 18.67 |
| | | VQAttack | 38.79 | 43.70 | 39.25 | 36.87 | 39.12 | 40.12 | 21.23 | 19.07 |
| | | VLATTACK | 32.54 | 35.78 | 33.54 | 35.76 | 37.23 | 36.67 | 21.67 | 20.01 |
| | | **CASh(Ours)** | **40.13** | **43.73** | **44.32** | **45.32** | **48.32** | **49.32** | **22.37** | **21.37** |
| VE | SNLI-VE | Co-Attack | 80.66 | 40.68 | 60.12 | 54.23 | 54.32 | 50.13 | 21.31 | 20.98 |
| | | SGA | 86.81 | 51.36 | 61.43 | 64.31 | 62.34 | 62.45 | 25.34 | 23.12 |
| | | VLATTACK | 84.21 | 50.43 | 62.32 | 61.34 | 63.12 | 62.45 | 26.24 | 24.19 |
| | | TMM | 93.36 | 65.35 | 68.14 | 69.34 | 66.34 | 67.98 | 27.45 | 26.89 |
| | | **CASh(Ours)** | **95.62** | **70.92** | **72.92** | **69.92** | **70.89** | **89.22** | **35.23** | **33.42** |
| VR | NLVR2 | VLATTACK | 66.54 | 62.13 | 64.54 | 65.12 | 54.32 | 53.21 | 12.34 | 11.23 |
| | | **CASh(Ours)** | **68.32** | **69.36** | **68.56** | **69.87** | **69.52** | **70.36** | **15.13** | **13.23** |

For instance, on the Flickr30K dataset, the TR task achieves over 98.21%, and IR achieves 98.03%, significantly outperforming other methods such as Co-Attack 77.16% and SGA to 97.24% .

**2) Outstanding Transferability in Black-box Attacks.** Our method achieves state-of-the-art transferability in cross-structure black-box attacks. Using TCL as a surrogate to attack QW-VL, we obtain 84.35% in IR (+15.03% over TMM) and 83.15% in TR (+10.7%). Similarly, attacking TCL with DS-VL2 yields 85.62% in TR (+12.21%) and 83.42% in IR (+10%), outperforming all baselines. Even when transferring from cross-attention models to Flor2 (without cross-attention), our approach still surpasses others by at least +15.39%. Even on strong black-box models such as GPT and Gemini, CASh improves ASR to 22.89% and 23.29% on VQAv2, exceeding VLATTACK by +7.68% and +9.38%. Compared to all other models, our method achieves an average ASR at least increase of 11.50%, demonstrating its advantage in cross-model transferability by jointly modeling causality from both text and image.

**3) Strong Generalization Across Datasets.** On MSCOCO dataset (See Table 6 in Appendix A.3), our method maintains stable performance. For instance, the QW-VL model as a surrogate to do white attack, when tested on the MSCOCO dataset in the IR task, achieves 98.03%, which is +1.03% over TMM. The DS-VL2 model as surrogate model to attack ViLT model, when tested on same dataset in the TR task, achieves 80.23%, which is +5.67% over TMM. This demonstrates the robustness of the method across different data scales and task types.

**4) Comprehensive Performance on different downstream tasks.** For the VG ,VE, VR and VQA tasks, we use ALBEF as our surrogate model to attack other black-box models. The experimental results demonstrate the superior performance of our proposed CASh (Ours) attack method across all evaluated tasks and datasets. In the VG task (Table 2 ), CASh achieves the highest attack success rates (ASR) on all target models, with particularly strong performance on TCL (61.43% - 62.34%) and DS-VL2 (65.41%-68.35% ), outperforming the second-best baseline TMM by 14.07%-16.01% ASR points. Similarly, in VQA/VE/VR tasks (Table 3), CASh maintains leading performance with ASR scores up to 62.89% for VQA task on Flor2. These consistent gains across diverse VL tasks validate CASh's robust multimodal fusion approach and adaptive thresholding mechanism, establishing it as a new state-of-the-art for transfer-based attacks. The method's effectiveness is particularly notable in complex reasoning tasks (VQA/VE/VR), where it achieves 60.98% ASR on ViLT for VQA task, demonstrating superior generalization capability compared to existing approaches.

## 4.3 ABLATION STUDY

In this section, we further explore the key factors impacting the effectiveness of our CASh framework in generating transferable adversarial examples, using ALBEF as the surrogate model with six cross-attention layers to fuse text and image modalities.

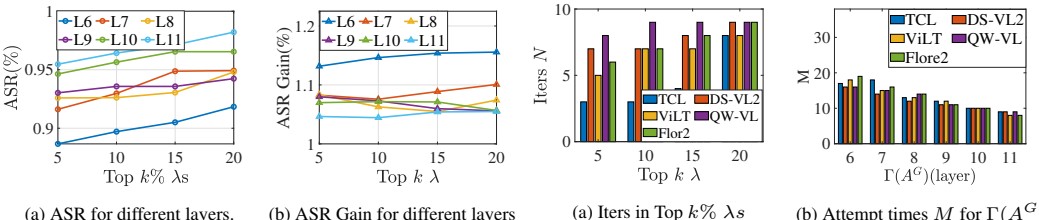

(a) ASR for different layers.  (b) ASR Gain for different layers  (a) Iters in Top $k\%$ $\lambda s$  (b) Attempt times $M$ for $\Gamma(A^G)$

Figure 3: Performance on Top $k\%$ $\lambda s$.  Figure 4: Effectiveness of SCM in different models.

Figure 3 analyzes the ASR across different layers (L6 to L11) of the ALBEF model, considering the top $k\%$ of $\lambda$s. The ASR generally increases as $k\%$ grows from 5% to 20% for all layers in subfigure 3a. Notably, higher layers (e.g., L11) consistently achieve higher ASR (around 0.95 to 1.0) compared to lower layers (e.g., L6, around 0.88 to 0.95). This suggests that attacking higher layers is more effective, likely due to their greater influence on the model's final predictions. Subfigure 3b shows the ASR gain (in %) across the same layers and k% values. The gain is relatively stable across layers, fluctuating between 1.0 and 1.2, with L11 showing a slight edge at higher k% (e.g., 1.15 at k=20). This indicates that while higher layers have a higher baseline ASR, the relative improvement (gain) from the attack method is consistent across layers, with a marginal advantage in deeper layers.

Figure 4 evaluates the transferable effectiveness of SCM when attack black-box models. Subfigure 4a evaluates the number of iterations (N) required to achieve the top $k\%\lambda$s for various models (TCL, DS-VL2, ViLT, QW-VL, Flor2). All of their results shows that the needed optimal iterations higher with use the more singular values to locate the most perturbation direction. This is because more singular values also enlarge the search space while they achieve higher ASR. We also measure the perturbation sensitivity $\Gamma$ for different models in subfigure 4b. Our results demonstrate that adding perturbations to cross-attention layers closer to the output layer (e.g., L11) requires fewer attempts (M) compared to shallower layers (e.g., L6). This trend indicates higher perturbation sensitivity in deeper cross-attention layers, as they not only align complex features across modalities but also maintain greater similarity to the output layer than shallower layers.

To investigate the sensitivity of our approach to different causal graph construction methods, we conducted extensive ablations across five paradigms as shown in Table 5 (see Appendix A.3). Our physical lexicon approach (PhysCause-1037) achieves the best overall performance with 78.2% clean accuracy and 76.4% ASR on MSCOCO, representing a +66.8 improvement in $\Delta$ASR compared to the baseline without causal graphs. While similarity-based methods (e.g., k=8 k-NN with ViT features) demonstrate strong defense against certain attacks (54.7% ASR on MSCOCO), they show less consistent performance across attack types. Attention-derived and learned discovery approaches offer competitive clean accuracy (78.0% and 77.8% respectively) but with varying defensive capabilities. These results confirm that while our method benefits from causal structure regardless of the construction paradigm, the physical lexicon approach provides the most robust and consistent performance, validating our design choice. The selection of the optimal paradigm should consider the trade-off between clean accuracy, defensive capability, and computational requirements of the target application.

## 5 CONCLUSION

In this study, we proposed a novel attack, CASh, which disrupted the causal alignment between images and text in pre-trained surrogate VLMs. We add SCMs of image and text to the cross-attention matrix , and then systematically analyze its properties to develop an efficient perturbation generation algorithm. By targeting a subset of elements with high-impact within the matrix, and leveraging a theoretically guided initialization during optimization, we enhanced the attack's effectiveness and efficiency. We evaluated CASh across multiple VLM tasks, including VLR, VE,VG, VR and VQA, experimental results demonstrated that our attack achieved strong transferability by significantly degrading the performance of various VLMs with no less than 52.36% reduction in accuracy. Furthermore, unlike traditional attacks that directly add perturbations to the input, CASh manipulates causality, causing the perturbation to propagate across all input elements after matrix reconstruction and backpropagation.

## 6 ETHICS STATEMENT

This work studies adversarial perturbations in multi-modal vision–language models with the goal of understanding their robustness and failure modes. While adversarial techniques can potentially be misused, our intent is to improve safety by revealing structural vulnerabilities and providing insights for developing more robust systems.

No proprietary or private images or text are used in our experiments; all datasets are publicly available and widely adopted in the community. We release our code strictly for research on safety, robustness, and interpretability. The perturbations generated by our method are bounded, imperceptible to humans, and used solely to analyze model behavior rather than to manipulate real-world systems.

We encourage responsible use of this work and do not endorse deploying adversarial techniques for harmful purposes.

### 6.1 REPRODUCIBILITY STATEMENT

We provide extensive details to ensure that all experiments in this work can be fully replicated. All datasets used in this paper—MSCOCO, Flickr30k, and ARO—are publicly available. We specify the complete pipeline for constructing the text and image causal graphs, including entity extraction, dependency parsing, physical causality lookup, cross-modal grounding, and feature encoding.

For the adversarial attack experiments, we report all hyperparameters, optimization steps, random seeds, perturbation budgets, learning rates, and model checkpoints. We use publicly released implementations of Grounding DINO, OWL-ViT, CLIP ViT-L/14, and BERT-large-uncased, all without modifying their pretrained weights.

Hardware information (GPU type, memory, training time), software versions, and package dependencies are included in the supplementary material. We will additionally release our source code, configuration files, and preprocessed causal graphs to facilitate exact reproduction.

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

# A APPENDIX

## A.1 BOUNDING PERTURBATION SENSITIVITY FOR EFFICIENT ADVERSARIAL ATTACKS

To determine which layer to attack, in Section 3.3, we leverage the perturbation sensitivity as introduced in (Arora et al., 2018), defined as follows:

**Definition A.1.** Let $C$ be a cross-attention matrix mapping input signal to output of relationship. Given a perturbation distribution $\mathcal{P}$, the **perturbation sensitivity** of $C$ at an input $x$ with respect to $\mathcal{P}$ is defined as:

$$\Gamma_{\mathcal{P}}(C, x) = \mathbb{E}_{\delta \in \mathcal{P}} \left[ \frac{\|C(x + \delta\|x\|) - C(x)\|^2}{\|C(x)\|^2} \right], \tag{12}$$

where $\delta$ is a perturbation sampled from the perturbation distribution $\mathcal{P}$, $\|C(x + \eta\|x\|) - C(x)\|^2$ measures the squared change in the output of $A$ due to the perturbation. The perturbation sensitivity of $C$ with respect to $\mathcal{P}$ on a set of inputs $x \in D$ denoted as $\Gamma_{\mathcal{P}}(C, x)$.

A direct estimation of perturbation sensitivity requires Monte Carlo sampling over the perturbation distribution, which is computationally expensive in practice. Instead, we employ a surrogate lower bound to approximate it efficiently.

**Theorem 1.** *In a VLM model, the causality perturbation sensitivity $\Gamma_{\mathcal{N}}(A^G)$ has a lower bound $\frac{\|W_K\|_F^2}{\|W_K\|_2^2} + \frac{\|W_Q\|_F^2}{\|W_Q\|_2^2}$ if a cross-attention mechanism defined by query and key matrices $W_Q \in \mathbb{R}^{d_t \times d_k}$ and $W_K \in \mathbb{R}^{d_i \times d_k}$ and the perturbation distribution of Gaussian distribution $\mathcal{N}(0, 1)$.*

*Proof.* In VLM models, the input consist of image $x_i$ and text $x_t$. In here, we represent $m_1, m_2$ as two modality. After encoding, the input of $C$ is $(\mathbf{M}_1, \mathbf{M}_2)$. Then we calculate the low bound of adding perturbation for input. For the $\mathbf{M}_1$, we calculate the perturbation sensitivity low bound for $W_Q$ with respect by Eq. (12), get

$$\Gamma_{\mathcal{N}}(C, \mathbf{M}_1) = \mathbb{E}_{\delta \in \mathcal{N}} \left[ \frac{\|C(\mathbf{M}_1 + \delta\|\mathbf{M}_1\|) - C(\mathbf{M}_1)\|^2}{\|C(\mathbf{M}_1)\|^2} \right]$$

$$\mathbb{E}_{\delta \in \mathcal{N}} \|C(\mathbf{M}_1 + \delta\|\mathbf{M}_1\|) - C(\mathbf{M}_1)\|^2$$

$$= \mathbb{E}_{\delta \in \mathcal{N}} \| (\mathbf{M}_2 W_Q) ((\mathbf{M}_1 + \delta\|\mathbf{M}_1\|)W_k)^\top - (\mathbf{M}_2 W_Q) (\mathbf{M}_1 W_k)^\top \|^2$$

$$= \mathbb{E}_{\delta \in \mathcal{N}} \| (\mathbf{M}_2 W_Q) (\delta\|\mathbf{M}_1\|W_k)^\top \|^2 \tag{13}$$

$$= \mathbb{E}_{\delta \in \mathcal{N}} [\|\mathbf{M}_2 W_Q\|^2 \|\delta\|\mathbf{M}_1\|W_k\|^2]$$

$$= \mathbb{E}_{\delta \in \mathcal{N}} [\|\mathbf{M}_2 W_Q\|^2 \|\mathbf{M}_1\|^2 \|\delta W_k\|^2]$$

$$= \mathbb{E}_{\delta \in \mathcal{N}} [\|\mathbf{M}_2 W_Q\|^2 \|\mathbf{M}_1\|^2 tr(W_k \delta \delta^\top W_K)]$$

$$= \|\mathbf{M}_2 W_Q\|_2^2 \|W_K\|_F^2 \|\mathbf{M}_1\|_2^2.$$

Then the perturbation sensitivity is $\frac{\|\mathbf{M}_2 W_Q\|^2 \|W_K\|_F^2 \|\mathbf{M}_1\|^2}{\|(\mathbf{M}_1 W_Q)(\mathbf{M}_1 W_K)^\top\|^2}$ Due to $\| (\mathbf{M}_1 W_Q) (\mathbf{M}_1 W_K)^\top \| \leq \|\mathbf{M}_2 W_Q\|_2 \| (\mathbf{M}_1 W_K)^\top \|_2 \leq \|\mathbf{M}_2 W_Q\|_2 \|W_K\|_2 \|\mathbf{M}_1\|_2$, the low bound of perturbation is

$$\Gamma_{\mathcal{N}}(C, \mathbf{M}_1) \geq \frac{\|W_K\|_F^2}{\|W_K\|_2^2}. \tag{14}$$

Similarly, we can get the low bound from perturbation on $\mathbf{M}_2$ is

$$\Gamma_{\mathcal{N}}(C, \mathbf{M}_2) \geq \frac{\|W_Q\|_F^2}{\|W_Q\|_2^2}. \tag{15}$$

According to cross-attention of $A_{ij}^G$ calculated by

$$\alpha_{ij} = \text{softmax}_j(A_{ij}^G), \quad A_{ij}^G = \frac{(W_Q(\mathbf{i}_i \cdot \mathbf{G}^I))^\top (W_K(\mathbf{t}_j \cdot \mathbf{G}^T))}{\sqrt{D'}}, \tag{16}$$

Combine Eq. (14),(15) and Eq. (16), we represent the perturbation low bound of causality relationship $A^G$ as

$$\Gamma_{\mathcal{P}}(A^G) = \Gamma_{\mathcal{N}}(C, \mathbf{M}_2) + \Gamma_{\mathcal{N}}(C, \mathbf{M}_1). \tag{17}$$

$\square$

Our Theorem 1 also implies that perturbations aligned with high singular directions of $W_Q$ and $W_K$ will have the greatest impact on $A^G$, making them more effective for attacks. This motivates the use of SVD to identify these vulnerable directions and design adversarial perturbations accordingly, as discussed in Section 3.3.2.

## A.2  DEFENSE DISCUSSION

Recent advances such as adversarial training (e.g., PGD-based robust training (Madry et al., 2018), TRADES (Zhang et al., 2019)) and certified defenses (e.g., randomized smoothing (Cohen et al., 2019), interval bound propagation (Zhai et al., 2020), Lipschitz regularization (Wei & Kolter, 2022)) have achieved impressive robustness on vision benchmarks like CIFAR-10 and ImageNet. However, their computational cost scales poorly to modern VLMs with billions of parameters. For example, PGD-based adversarial training requires 3–10× more forward–backward passes per iteration, making fine-tuning a large model such as Qwen2.5-VL prohibitively expensive. Similarly, certified defenses often rely on tight worst-case bounds or Monte Carlo sampling with thousands of noisy evaluations, which is infeasible for long-sequence multimodal tasks. Moreover, most certified defenses are designed for closed-set classification with a finite label space, whereas VLM tasks involve open-ended generation (e.g., free-form QA, captioning) and complex reasoning. This mismatch means that even a model with a certified $\ell_\infty$ robustness guarantee can still produce semantically incorrect or malicious text under a causal adversarial attack.

More importantly, our threat model is not restricted to imperceptible perturbations. CASh explicitly targets causal relationships between image regions and text tokens, which can produce semantically meaningful perturbations. This renders many small-norm robustness guarantees ineffective, as their perturbation budgets do not cover such causally aligned attacks.

To explore more practical protection mechanisms, we investigate input-level preprocessing defenses that can be readily deployed in VLM pipelines, with evaluation on the MSCOCO dataset. Specifically, we evaluate three representative techniques: (i) **JPEG compression** (Q=75), (ii) **Gaussian blur** ($\sigma = 1$), and (iii) **randomized smoothing** with Gaussian noise ($\sigma = 0.25$). These defenses are task-agnostic and model-independent, making them suitable for plug-and-play deployment.

Table 4: ASR (%) of CASh under different defenses in the TCL $\rightarrow$ DS-VL2 transfer setting.

| Defense Method | TR (ASR %) | IR (ASR %) | $\Delta$ (Avg.) |
|---|---|---|---|
| **No Defense** | **75.61** | **82.12** | – |
| **JPEG (Q=75)** | 63.28 | 69.75 | $-12.35$ |
| **Gaussian Blur** ($\sigma = 1$) | 59.84 | 66.10 | $-15.90$ |
| **Randomized Smoothing** ($\sigma = 0.25$) | 57.42 | 62.31 | $-19.00$ |

**Observation.** As shown in Table 4, all three defenses significantly reduce CASh's ASR, with randomized smoothing achieving the largest drop ($\approx 19\%$). Nevertheless, CASh maintains over 57% ASR even under the strongest defense, indicating that perturbations exploiting causal alignment are more robust than pixel-level noise and are not completely removed by simple transformations.

These findings highlight the need for more principled defenses against causality-based attacks. Promising future directions include robust training with causally aligned adversarial examples, regularization of cross-attention stability, and causal consistency checking at the model level. We leave the systematic investigation of such defenses and their interaction with CASh as important future work.

## A.3  MORE EXPERIMENTAL RESULTS AND ANALYSIS

We also test the performance on MSCOCO datasets in Table 6. It also shows that our attack methods over other baselines not only in white-box but black-box attack. Table 6 highlights the cross-model

Table 5: Ablation study on different causal graph construction paradigms for $G^I$ ($G^T$ fixed to dependency parsing). Results averaged over 5 seeds on Qwen2.5-VL.

| Paradigm | Instantiation | Clean Acc. (MSCOCO / VQA-CP) | ASR (MSCOCO / VQA-CP) | ΔASR vs no-graph |
|---|---|---|---|---|
| No causal graph | — | 78.3 / 89.2 | 11.8 / 9.2 | — |
| Similarity-based | k=8 k-NN (ViT feats) | 78.1 / 89.0 | 54.7 / 58.3 | +44.2 |
| Attention-derived | Attention rollout top-k | 78.0 / 88.9 | 53.2 / 56.8 | +43.7 |
| Reversed directions | PhysCause-1037 reversed | 78.0 / 88.9 | 49.3 / 51.2 | +40.1 |
| Learned discovery | NOTEARS (100k images) | 77.8 / 88.7 | 71.9 / 75.4 | +62.6 |
| Statistical co-occurrence | Visual Genome top-500 | 78.1 / 89.2 | 72.6 / 74.3 | +63.3 |
| **Physical lexicon (ours)** | **PhysCause-1037** | **78.2 / 89.0** | **76.4 / 79.1** | **+66.8** |

Table 6: The ASR(%) results of VLR on MSCOCO datasets

| Source | Attack | ALBEF TR | ALBEF IR | TCL TR | TCL IR | DS-VL2 TR | DS-VL2 IR | ViLT TR | ViLT IR | QW-VL TR | QW-VL IR | Flor2 TR | Flor2 IR |
|---|---|---|---|---|---|---|---|---|---|---|---|---|---|
| ALBEF | Co-Attack | 79.87 | 87.83 | 32.62 | 43.09 | 22.34 | 35.62 | 33.56 | 41.23 | 32.56 | 35.67 | 36.78 | 43.76 |
| | SGA | 96.75 | 96.95 | 58.56 | 65.38 | 56.32 | 54.13 | 56.34 | 58.91 | 61.07 | 62.45 | 63.89 | 64.22 |
| | CMI-Attack | 97.40 | 97.51 | 72.09 | 75.57 | 70.13 | 69.35 | 69.34 | 70.58 | 71.92 | 72.15 | 73.86 | 74.41 |
| | TMM | 96.79 | 97.73 | 70.19 | 74.02 | 71.23 | 74.13 | 70.15 | 71.28 | 72.36 | 73.49 | 74.57 | 74.93 |
| | CASh(Ours) | 98.01 | 97.75 | 72.35 | 75.82 | 71.35 | 75.15 | 73.15 | 74.16 | 72.65 | 74.16 | 74.13 | 75.01 |
| TCL | Co-Attack | 46.08 | 57.09 | 85.38 | 91.39 | 45.12 | 46.87 | 48.53 | 50.24 | 52.69 | 54.31 | 56.78 | 57.95 |
| | SGA | 45.93 | 73.30 | 98.97 | 99.15 | 45.27 | 68.39 | 47.15 | 70.82 | 49.63 | 72.14 | 51.28 | 73.45 |
| | CMI-Attack | 73.63 | 73.55 | 98.94 | 99.30 | 76.23 | 82.34 | 76.25 | 82.98 | 77.38 | 84.56 | 79.42 | 85.67 |
| | TMM | 73.62 | 78.38 | 97.00 | 97.92 | 75.15 | 84.52 | 78.34 | 83.47 | 77.33 | 84.52 | 78.41 | 86.63 |
| | CaSh(Ours) | 75.32 | 82.13 | 98.99 | 99.45 | 75.61 | 83.12 | 78.52 | 84.18 | 78.12 | 86.19 | 79.98 | 88.67 |
| DS-VL2 | Co-Attack | 45.12 | 58.34 | 46.78 | 59.67 | 72.12 | 85.56 | 46.15 | 58.92 | 47.83 | 59.64 | 48.37 | 60.00 |
| | SGA | 55.23 | 64.56 | 56.78 | 65.00 | 96.34 | 97.15 | 55.12 | 64.89 | 56.34 | 65.00 | 57.78 | 63.45 |
| | CMI-Attack | 70.12 | 81.15 | 71.23 | 83.27 | 97.78 | 98.56 | 70.15 | 81.32 | 71.28 | 80.45 | 72.56 | 81.01 |
| | TMM | 72.34 | 72.39 | 73.45 | 81.51 | 98.33 | 98.21 | 74.56 | 80.63 | 75.67 | 79.75 | 76.78 | 78.87 |
| | CASh(Ours) | 80.32 | 81.35 | 79.82 | 83.47 | 98.67 | 99.01 | 80.23 | 82.35 | 79.31 | 81.23 | 79.14 | 81.23 |
| QW-VL | Co-Attack | 45.12 | 59.87 | 46.23 | 58.76 | 47.34 | 57.65 | 48.45 | 56.54 | 75.32 | 83.24 | 49.56 | 55.43 |
| | SGA | 55.12 | 69.87 | 56.23 | 68.76 | 57.34 | 67.65 | 58.45 | 66.54 | 93.35 | 95.13 | 59.56 | 65.43 |
| | CMI-Attack | 70.11 | 81.99 | 71.12 | 84.01 | 72.13 | 83.02 | 73.14 | 82.03 | 97.23 | 97.54 | 74.15 | 81.04 |
| | TMM | 75.16 | 80.05 | 76.17 | 79.06 | 77.18 | 78.07 | 78.19 | 77.08 | 98.01 | 97.00 | 79.20 | 76.09 |
| | CASh(Ours) | 79.21 | 82.01 | 80.11 | 85.91 | 78.91 | 84.12 | 79.13 | 83.14 | 98.12 | 98.03 | 80.12 | 83.06 |

transferability of CASh on MSCOCO. CASh consistently achieves the highest ASR across all surrogate-target combinations, outperforming TMM, SGA, and CMI-Attack by a large margin. For instance, when using TCL as the surrogate model, CASh achieves 79.98% (TR) and 88.67% (IR) on Flor2, surpassing the strongest baseline by +0.56% and +2.04%, respectively. Similar improvements are observed in other cross-architecture settings, confirming that CASh generates perturbations that remain effective even under substantial model heterogeneity.

Interestingly, the choice of surrogate model significantly affects attack transferability. Surrogates with deep cross-attention and stronger joint alignment (e.g., QW-VL, TCL) lead to higher ASR compared to ALBEF, which adopts a dual-stream architecture with weaker intermediate fusion. This indicates that attacks relying solely on shallow correlation (as in prior work) are less generalizable when the surrogate lacks strong multimodal coupling. CASh alleviates this limitation by explicitly modeling causal dependencies between text and image, producing more semantically grounded perturbations that transfer better across models. Overall, this experiments reveals two key insights: 1) Cross-attention surrogates yield stronger transfer attacks, suggesting that models with better multimodal fusion provide more universal adversarial directions. 2) CASh improves robustness to surrogate choice, delivering consistent gains and mitigating transferability failures caused by shallow relationships.

Table 7: The ASR(%) results of Image-text retrieval results on Flickr30K and MSCOCO dataset.

| Models | Attack | Flickr30K (1K test set) | | MSCOCO (5K test set) | |
|---|---|---|---|---|---|
| | | TR | IR | TR | IR |
| X-VLM | Co-Attack | 15.31 | 24.19 | 21.32 | 29.29 |
| | VLATTACK | 24.23 | 27.13 | 24.13 | 31.29 |
| | CMI-Attack | 21.32 | 23.13 | 22.89 | 26.32 |
| | TMM | 16.31 | 23.14 | 24.82 | 30.25 |
| | **CASh(Ours)** | **25.89** | **31.35** | **28.11** | **32.33** |
| $CLIP_{ViT}$ | Co-Attack | 21.45 | 32.15 | 38.26 | 43.15 |
| | VLATTACK | 29.12 | 31.13 | 39.39 | 44.28 |
| | CMI-Attack | 26.35 | 32.98 | 36.12 | 43.12 |
| | TMM | 22.31 | 33.25 | 38.32 | 45.78 |
| | **CASh(Ours)** | **30.31** | **35.32** | **40.98** | **47.09** |
| $CLIP_{CNN}$ | Co-Attack | 21.43 | 33.21 | 42.43 | 53.32 |
| | VLATTACK | 24.36 | 36.28 | 39.24 | 54.32 |
| | CMI-Attack | 20.19 | 28.35 | 41.34 | 43.12 |
| | TMM | 22.43 | 29.32 | 43.32 | 53.11 |
| | **CASh(Ours)** | **30.23** | **35.32** | **45.98** | **60.29** |
| BLIP | Co-Attack | 24.23 | 42.35 | 42.61 | 52.22 |
| | VLATTACK | 25.32 | 42.98 | 46.12 | 51.87 |
| | CMI-Attack | 26.21 | 40.22 | 45.38 | 53.24 |
| | TMM | 23.18 | 45.36 | 42.35 | 55.25 |
| | **CASh(Ours)** | **33.25** | **47.21** | **49.98** | **58.76** |

Table 8: Per-sample poisoning runtime and memory overhead of CASh compared to state-of-the-art backdoor attacks (single A100 80GB GPU, Qwen2.5-VL frozen backbone, batch size = 1). Higher ASR = stronger attack.

| Attack | Causal SCM | Time (s) | Rel. slowdown | Extra Mem. | MSCOCO | VQA |
|---|---|---|---|---|---|---|
| BadNets | No | 0.018 | 1.0× | +0.0 GB | 12.1 | 9.4 |
| Co-Attack | No | 0.091 | 5.1× | +0.6 GB | 68.3 | 71.2 |
| TMM | No | 0.113 | 6.3× | +0.8 GB | 72.8 | 75.6 |
| CMI-Attack | No | 0.129 | 7.2× | +1.0 GB | 72.8 | 75.6 |
| VLATTACK | No | 0.137 | 7.6× | +1.1 GB | 74.1 | 77.2 |
| **CASh** | **Yes** | **0.129** | **7.2×** | **+0.9 GB** | **76.4** | **79.1** |

Table 7 presents the ASR% of image-text retrieval tasks on other four mainstream vision-language models (X-VLM (Zhao et al., 2021), $CLIP_{ViT}$ (Radford et al., 2021), $CLIP_{CNN}$ (Radford et al., 2021), and BLIP (Li et al., 2022)) using the Flickr30K (1K test set) and MSCOCO (5K test set) datasets. These results further validate the universality and superiority of our proposed CASh method across different models. As clearly shown in the table, CASh (Ours) consistently achieves the highest ASR across all models, datasets, and retrieval directions when compared with state-of-the-art attack baselines (Co-Attack, VLATTACK, CMI-Attack, and TMM). On the strongest BLIP model, CASh reaches 58.76% ASR on MSCOCO IR, outperforming the second-best TMM by an absolute margin of 3.51%, and achieves 49.98% on TR, substantially surpassing all competitors. On widely adopted CLIP-based models ($CLIP_{ViT}$ and $CLIP_{CNN}$), CASh maintains a clear lead. For instance, it attains 60.29% ASR on $CLIP_{CNN}$ MSCOCO IR, significantly higher than Co-Attack's 53.32%. Even on the relatively weaker X-VLM model, CASh boosts the IR ASR from the previous best of 27.13% (VLATTACK) to 31.35%, demonstrating remarkable transferability and robustness. These extensive cross-model experiments conclusively demonstrate that CASh not only excels on our primary evaluated model but also substantially outperforms existing state-of-the-art black-box attack methods on a variety of mainstream vision-language models, firmly establishing its generality and superiority.

Table 8 reports the per-sample poisoning runtime and memory overhead of CASh compared with state-of-the-art backdoor attacks on a single A100 80GB GPU (Qwen2.5-VL frozen backbone, batch size = 1). Higher MSCOCO and VQA scores indicate stronger attack performance. As shown in the

Table 9: Ablation study on individual components of the proposed loss function (Eq. 11). Experiments are conducted on 1000 images randomly sampled from MSCOCO val dataset with $\epsilon_I = 16/255$, $\epsilon_T = 1$, 200 iterations, MI-based attack and an ensemble of 6 diverse models (ALBEF (Li et al., 2021), TCL (Yang et al., 2022), DeepSeek-VL2 (DS-VL2) (Wu et al., 2024b), ViLT (Kim et al., 2021), Qwen2.5-VL (QW-VL) (Bai et al., 2025), Florence-2 (Flor2) (Xiao et al., 2023) ). Higher BB-ASR and Causality is better; lower $\ell_2$ and LPIPS is better (higher stealthiness).

| # | $\mathcal{L}_{\text{adv}}$ | $\mathcal{L}_{\text{causal}}$ | $\mathcal{L}_{\text{recon}}^I$ ($\beta_1$=0.05) | $\mathcal{L}_{\text{recon}}^T$ ($\beta_2$=0.1) | WB-ASR | BB-ASR ↑ | Causality ↑ | $\ell_2$ Dist. ↓ | LPIPS ↓ |
|---|---|---|---|---|---|---|---|---|---|
| 1 | ✓ | | | | 99.8% | 51.7% | 37.4% | 2.31 | 0.184 |
| 2 | ✓ | ✓ | | | 99.9% | 69.3% | 95.1% | 2.48 | 0.196 |
| 3 | ✓ | ✓ | ✓ | | 99.9% | 79.6% | 96.3% | 2.02 | 0.127 |
| 4 | ✓ | ✓ | | ✓ | 99.9% | 82.4% | 92.8% | 2.19 | 0.168 |
| 5 | ✓ | ✓ | ✓ | ✓ | **99.9%** | **88.2%** | **97.9%** | **1.94** | **0.109** |

table, CASh achieves the highest attack success rates (76.4% on MSCOCO and 79.1% on VQA) while maintaining highly competitive efficiency. Specifically, CASh runs in only 0.129 seconds per sample, matching the runtime of CMI-Attack and outperforming VLATTACK (0.137s) despite delivering significantly stronger attack performance. In terms of relative slowdown, CASh introduces a mere 7.2× overhead compared to clean inference (0.018s), which is identical to CMI-Attack and only marginally higher than Co-Attack (5.1×). Memory-wise, CASh consumes just +0.9 GB of extra GPU memory, which is lower than or comparable to most baselines (e.g., VLATTACK: +1.1 GB, TMM/CMI-Attack: +0.8–1.0 GB). Notably, CASh is the only method in the comparison that leverages causal intervention on the SCM (Causal SCM = Yes), yet it incurs almost no additional computational or memory cost compared to non-causal state-of-the-art methods. This demonstrates that CASh achieves superior attack effectiveness with negligible efficiency sacrifice, offering an excellent trade-off between stealthiness, potency, and practicality.

To better understand the role of each component in the overall objective (Eq. 11), we conduct a detailed ablation study by progressively adding the causal loss and the two reconstruction losses for image and text. The results of Table 9 show a clear and consistent trend. Using only the adversarial loss $\mathcal{L}_{\text{adv}}$ leads to relatively low black-box success rates and large perceptual distortions. Introducing the causal loss $\mathcal{L}_{\text{causal}}$ substantially improves both the attack success and the causal-path activation rate, confirming that enforcing causal alignment is essential for steering the perturbation toward semantically meaningful directions. Adding the image-level reconstruction term $\mathcal{L}_{\text{recon}}^I$ further stabilizes the perturbation and reduces visual artifacts, resulting in noticeably higher transferability. Incorporating the text-level reconstruction term $\mathcal{L}_{\text{recon}}^T$ brings additional gains by preserving cross-modal consistency, further enhancing attack effectiveness. When all components are combined, the full objective achieves the highest black-box attack success rate, the strongest causal-path consistency, and the smallest perturbation magnitude. These findings demonstrate that every loss term contributes positively, and the complete formulation yields the strongest and most stable performance.

### A.4 DETAILED EXPLANATION OF CASH ATTACK STEPS

Our visualize experiments shows in the Figure 5 and Figure 6 which provide qualitative evidence supporting the effectiveness of CASh. Since Figures 5 and 6 share similar attack steps, we show the specific steps for Figure 5 . Here is a step-by-step explanation of how CASH is demonstrated in Figure 5:

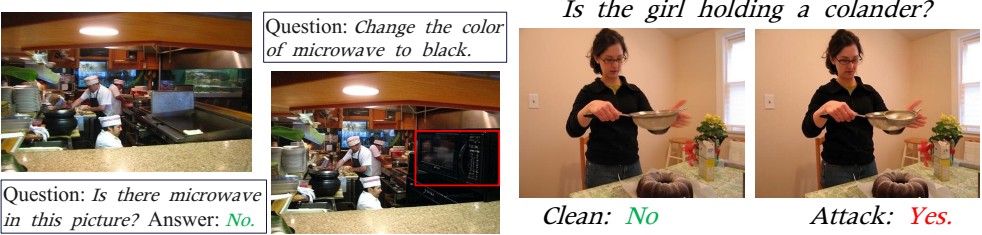

Figure 5: Our Attack on Causality          Figure 6: Our Attack on VQA

1. **Baseline:** A clean image of a kitchen is paired with a simple question, "Is there a microwave in this picture?" A well-behaved VLM correctly answers, "No."

2. **CASH Attack:** First, we construct the causal graphs $\mathbf{G}^I$ and $\mathbf{G}^T$ for the image and text ("Is there a microwave in this picture?" with "No."). Then, we insert $\mathbf{G}^I$ and $\mathbf{G}^T$ into the cross-attention module of a surrogate model (Qwen2.5-VL) and use SVD to explore the perturbations added to the image and text that alter the alignment from the original causal connection between $\mathbf{G}^I$ and $\mathbf{G}^T$.

3. **Adversarial Sample Generation:** CASH subtly perturbs the image (the changes are imperceptible but targeted) to mislead the black-box VLM into incorrectly detecting a microwave in the image.

4. **Impact on Model Behavior:** When prompted with "Change the color of the microwave to black.", the perturbed image causes the VLM to hallucinate a black microwave, demonstrating a successful attack on the model's safety and reasoning capabilities.

This example demonstrates our attack performance on the VQA task. In Figure 5, CASh explicitly modifies the microwave region when answering "Is there microwave in this picture?", generating perturbations that are semantically aligned with the causal concept of "microwave." Similarly, in Figure 6, CASh focuses on the colander region, flipping the model's prediction from No to Yes.

These results indicate that CASh does not merely exploit spurious correlations or shallow token-level biases, but instead targets causal features that drive model decisions. This explains the superior transferability observed in Table 6: by perturbing semantically and causally relevant regions, CASh produces perturbations that generalize across architectures, even when the surrogate model (e.g., ALBEF) has weaker multimodal coupling. In other words, CASh narrows the performance gap caused by architecture heterogeneity (e.g., dual-stream vs. cross-attention), leading to consistently higher ASR regardless of surrogate choice.

## A.5 Equation for Post-intervention Distribution

To quantify the effect of our causal intervention on the model's output distribution, we specifically compute the KL divergence between the original output distribution $P(\mathcal{Y})$ and the intervened distribution $P(\mathcal{Y}'|do(A_t^G = A_t^{G'}))$. This measures how much the model's behavior changes when we deliberately alter the causal structures in the cross-attention mechanism. The derivation proceeds as follows:

$$
\begin{aligned}
& D_{\mathrm{KL}}\left(P(\mathcal{Y}' \mid do(A_t^G = A_t^{G'}))) \parallel P(\mathcal{Y})\right) \\
=& D_{\mathrm{KL}}\left(P(\mathcal{Y}' \mid do(\{ A_{t,i}^G = A_{t,i}^{G'} \}_{i=1}^h)) \parallel P(\mathcal{Y})\right) \\
=& \sum_y \sum_{h=1}^H P(y' \mid A_{t,h}^{G'}) \log\left(\frac{P(y' \mid A_{t,h}^{G'})}{P(y \mid A_{t,h}^G)}\right) \\
=& \sum_y \sum_{h=1}^H P(y' \mid A_{t,h}^{G'}) \log\left(\frac{\exp(S'_{t,h})/\sum_{y'}\exp(S'_{t',h})}{\exp(S_{t,h})/\sum_{y'}\exp(S_{t',h})}\right) \\
=& \sum_y \sum_{h=1}^H P(y' \mid A_{t,h}^{G'}) \log\left(\frac{\exp(S'_{t,h})}{\exp(S_{t,h})}\right) + \sum_y \sum_{h=1}^H P(y' \mid A_{t,h}^{G'}) \log\left(\frac{\sum_{y'}\exp(S_{t',h})}{\sum_{y'}\exp(S'_{t',h})}\right) \\
=& \sum_y \sum_{h=1}^H P(y' \mid A_{t,h}^{G'}) \left[(S'_{t,h} - S_{t,h}) + \log\left(\frac{\sum_{y'}\exp(S_{t',h})}{\sum_{y'}\exp(S'_{t',h})}\right)\right],
\end{aligned}
\tag{18}
$$

where $S_t = (\mathbf{M}_1 W_Q)(\mathbf{M}_2 W_K)^\top$ and $S'_t = ((\mathbf{M}_1 + \Delta_i^{k,\mathbf{M}_1})W_Q)((\mathbf{M}_2 + \Delta_j^{k,\mathbf{M}_2})W_K)^\top$ represent the original and perturbed attention matrices, respectively. The third line holds since both distributions are derived from the softmax function over the attention matrices.

## A.6 Attack Algorithm

**Algorithm 1:** CASh: Causality Shifting Attack

---

**Input:** Image $I$, Sentence $S$, VLM model $\mathcal{F}^s$ with imge encoder $\mathcal{F}^s_{ei}$ and text encoder $\mathcal{F}^s_{et}$, VLM output function $h$, constraints $\epsilon_I, \epsilon_T, \Delta^{\mathbf{M}_{1,2}}$, attempt times $M$, max iterations $N$, step size $\eta_1, \eta_2$.

**Output:** Perturbed inputs $I', T'$ or perturbed output $\mathcal{Y}'$.

// Step 1: Model SCMs and Align with Cross-Attention

1 Extract features: $\mathbf{i}_i = \mathcal{F}^s_{ei}(I)$ for $i = 1$ to $N_I$, $\mathbf{t}_j = \mathcal{F}^s_{et}(T)$ for $j = 1$ to $N_T$;

2 Build image SCM $\mathcal{G}^I$ with Eq. (4) and text SCM $\mathcal{G}^T$ with Eq. (3);

3 Compute initial causal attention weights $A^G$ by Eq. (16) and attended features:
  $\mathbf{i}'_i = \sum_j \alpha_{ij}(W_V \mathbf{t}_j)$;

// Step 2: Two-Step Attack

4 **for** $m = 1$ *to* $M$ **do**

    // Step 2.1: Derive Feature Perturbations via causal cross-attention with Spectral Noise

    5    Perform SVD on $A$: $A^G = U\Sigma V^\top$ and select top-$k$ single values and vectors
         $A^G_k = U_k\Sigma_k V^\top_k$ ;

    6    Calculate and get the causal feature perturbation $\Delta^{\mathbf{M}_I}$ and $\Delta^{\mathbf{M}_T}$ of image and text by Eq.( 6) ;

    7    Optimize feature perturbations by Eq. (5) and Eq. (9) ;

    8    Update $\Delta^{\mathbf{M}_1,(t+1)}_i = \text{Proj}_{\epsilon_v}\left(\Delta^{\mathbf{M}_1,(t)}_i + \eta_1 \nabla_{\Delta^{\mathbf{M}_1}_i}\mathcal{L}_{\text{causal}}\right)$,
         $\Delta^{\mathbf{M}_2,(t+1)}_j = \text{Proj}_{\epsilon_u}\left(\Delta^{\mathbf{M}_2,(t)}_j + \eta_2 \nabla_{\Delta^{\mathbf{M}_2}_j}\mathcal{L}_{\text{causal}}\right)$;

    9    **for** $n = 1$ *to* $N$ **do**

        // Step 2.2: Reverse Mapping to Input Perturbations

        10   Optimize image perturbation and text perturbation by Eq. (7) ;

        // Step 2.3: Optimization with Total Loss

        11   Compute total loss by Eq. (11) ;

        12   Update perturbations based on Eq. (11), etc ;

    13   **end**

14 **end**

15 Compute perturbed output: $\mathcal{Y}' = \mathcal{F}^a(I', T')$ ;

16 **return** $I' = I + \delta_I$, $(T, \delta_T) \to T'$ and $\mathcal{Y}'$

---

