# OpenReview forum: "CASh: Causality Alignment Shifting to Unveil Vulnerabilities in Visual-Language Model"
_ICLR.cc/2026/Conference — ICLR 2026 Conference Desk Rejected Submission_

### Official Review · Reviewer_4Ybc · 2025-10-24

**Soundness:** 2
**Presentation:** 1
**Contribution:** 2
**Rating:** 2
**Confidence:** 2

**Summary:**

This paper introduces CASh (Causality Alignment Shifting), a causality-based adversarial attack framework for VLMs. CASh perturbs the latent causal relationships between image and text modalities by modeling them through SCMs and analyzing their cross-attention alignments. Using SVD they identify high-impact causal directions. Experiments across various VLM tasks and several models show that CASh consistently achieves higher attack success and improves transferability by over 20% compared to prior methods.

**Strengths:**

The causality-driven adversarial attack framework represents a clear conceptual novelty beyond pixel-level attacks on VLMs. I like it because it's more explainable and intuitive compared to previous adversarial attacks.
The use of SVD to identify high-impact causal directions is also an interesting methodological choice and seems to be pretty effective. Overall, I think the motivation behind this paper is interesting; however, it needs significant rewriting and modifications as mentioned in the weaknesses section.

**Weaknesses:**

- Respectfully, the writing quality of the paper is very poor and appears to have been prepared hastily without sufficient proofreading. There are numerous grammatical and logical errors that make the paper difficult to read, unenjoyable, and at times confusing. Below are only a few examples, though many more exist:
visual Reasoning(VR)Chen et al. (2023b) → missing space.
as SGA Lu et al. (2023), Co-Attack Zhang et al. (2022a), VLATTACK Yin et al. (2024a), and TMM Wang et al. (2024a), → inconsistent and poor citation style.
a robust means to test VLM resilience Lu et al. (2023) → same citation issue.
For example, Figure 1 shows that the “kitchen” and “microwave” form a causal relationship... → multiple grammatical errors and incorrect verb agreement.

- Beyond grammar and typography, the overall presentation and writing logic are of low quality. The Introduction contains many long, convoluted sentences and fails to clearly present the motivation, problem, importance, and proposed solution. The section reads as disorganized and hard to follow, requiring significant rewriting. The excessive use of em dashes (—) also suggests possible overreliance on LLM-generated text.

- Section 2.3 on causal adversarial attacks is poorly structured. Each related work is briefly summarized without explaining how they connect to one another or to this paper’s contributions. As a result, the section reads like a collection of disconnected summaries rather than a coherent narrative situating prior work in context.

- Shallow treatment of SCMs, which are central to the proposed method, is also a weakness that makes the methodology less clear. The paper does not clearly explain how these causal graphs are constructed, validated, or whether they accurately capture the model’s underlying causal dependencies.
Moreover, several methodological details such as the implementation of causal adjacency matrices, SVD component selection, and reconstruction losses, are vague or missing, making the approach difficult to reproduce. A deeper and more transparent investigation into SCM background, validation, and sensitivity would have substantially strengthened the paper’s technical rigor and credibility.

- It would have been much better if the paper had included Reproducibility and Ethics Statement sections, as strongly encouraged by ICLR, given that this work focuses on adversarial attacks and involves extensive experimental setups.

**Questions:**

Please refer to the weaknesses.

---

> ### Author Response · Authors · 2025-11-21
> **Response to Reviewer 4Ybc (Part 1/2)**
>
> We sincerely thank the reviewer for the valuable comments and suggestions. Below, we provide detailed responses to each point and have revised the manuscript accordingly.
>
> > **Weakness1:** The paper’s writing quality is poor, with numerous grammatical, logical, and citation-formatting issues that make it hard to read and follow.
>
>  We sincerely thank the reviewer for the detailed and constructive feedback. We acknowledge the issues raised regarding writing quality, structure, and methodological clarity. We have substantially revised the paper to address all points.  We agree with the reviewer that the previous draft contained grammatical errors, inconsistent citation formatting, and typographical mistakes. We have now conducted a comprehensive rewrite and professional proofreading of the entire paper. Specifically:
> - We have corrected all grammatical issues, spacing mistakes (e.g., “visual Reasoning(VR)Chen”), and verb-agreement errors.
> - We have standardized the citation formatting to ensure consistency across all references, changing from “\cite{}” to “\citep{}”.
> - We have rewritten Sentences around Figure 1 for clarity and correctness.
> We appreciate the reviewer’s attention to detail and have ensured that the revised paper reads clearly and professionally.
>
> > **Weakness 2:** The paper’s overall presentation and logic are unclear and disorganized, especially in the Introduction, and that the writing quality is low enough to require substantial rewriting.
>
> We appreciate the reviewer’s concern regarding the clarity and structure of the Introduction. In the revised version (See red color in Introduction section), we have:
> - Rewritten the Introduction to clearly articulate the motivation, problem formulation, significance, and our key contributions in a concise and structured manner.
> - Removed overly long or convoluted sentences.
> - Reduced the usage of em dashes and improved overall readability.
> The Introduction now offers a much clearer and more coherent narrative.
>
> > **Weakness 3:** Section 2.3 is weakly organized, offering disconnected summaries of prior work without explaining their relationships or how they relate to this paper’s contributions.
>
> We appreciate the reviewer’s helpful feedback regarding the structure of Section 2.3. We agree that the original version read as a set of disconnected summaries without clearly articulating how prior works relate to one another or to our contribution. In the revision, we have reorganized this section around a coherent conceptual narrative.
> Specifically, we now
>
> (1) group existing causality-based adversarial attacks into three methodological categories: distribution-level causal alignment, counterfactual-optimization approaches, and attacks targeting causal inference pipelines;
>
> (2) clearly explain the connections and distinctions among these research lines;
>
> (3) explicitly position our method as a complementary direction that exploits causal misalignment in multimodal cross-attention rather than preserving causal structure. This restructuring clarifies the role of prior work and situates our contribution within the broader landscape. The revised section (included in the camera-ready submission) provides a more cohesive and informative overview consistent with the reviewer’s suggestion.

---

> ### Author Response · Authors · 2025-11-21
> **Response to Reviewer 4Ybc (Part 2/2)**
>
> >**Weakness 4:** The paper provides an incomplete and unclear treatment of SCMs and key methodological details, making the proposed approach difficult to understand and reproduce.
>
> Thank you for this insightful comment. We appreciate the opportunity to clarify this part of the method.
>
> In the revised manuscript, we have substantially rewritten Section 3.1 to provide a clearer and more rigorous description of how the text and image causal graphs $\mathcal{G}^T$ and $\mathcal{G}^I$ are constructed ( see subsection 3.1.1 and subsection 3.1.2 ). Specifically, the updated section now includes:
> - **Extraction of textual nodes.** We describe in detail how full noun phrases and named entities are identified from the caption, how each entity is represented through both its surface span and its head lemma, and how these representations are used for feature encoding and later causal reasoning.
> - **Construction of the text causal graph.** We now explain the process for deriving linguistic causal relations from dependency parsing, as well as how physical commonsense relations from the PhysCause lexicon are incorporated. The final adjacency matrix is defined as the union of these syntactic and physical causal edges.
> - **Construction of the image causal graph.** We clarify how image entities are obtained using Grounding DINO guided by the canonical text labels, how visual features are extracted using CLIP ViT-L/14, and how the image-side causal structure is aligned with the text-side graph to ensure consistent semantics across modalities.
>
> We hope these clarifications make the construction process of both causal graphs more transparent and adequately address the reviewer’s concern.We are happy to provide any further details or examples the reviewer requests.
>
> >**Weakness 5:** It would have been much better if the paper had included Reproducibility and Ethics Statement sections.
>
> Thank you very much for raising this important point. We fully agree that including both a Reproducibility Statement and an Ethics Statement will strengthen the paper, especially given the adversarial nature of our work and the scale of the experimental setup.
>
> **Reproducibility.**
>  In the revised manuscript, we have added a dedicated Reproducibility section following the ICLR guidelines. This section will summarize all implementation details necessary to replicate our results, including dataset usage, causal graph construction procedures, model configurations, hyperparameters, optimization settings, and evaluation protocols. To further support reproducibility, we will release the full codebase, preprocessing scripts, and configuration files at the camera-ready stage. These materials will allow researchers to reproduce our experiments without difficulty.
>
> **Ethics Statement.**
>  We also have added an Ethics Statement that discusses the implications of adversarial attacks on multimodal systems, the intended research-oriented use of our method, the potential risks, and the broader benefits for robustness and safety research. We will clarify that all datasets are publicly available and that the adversarial perturbations are designed for analytical purposes rather than deployment.
>
> We sincerely appreciate this constructive suggestion, and we believe that incorporating these two sections will significantly improve the completeness and clarity of the paper.
>
> **Summary:** We are truly grateful for the reviewer’s candid and rigorous assessment of the paper’s presentation and structure. This feedback was a necessary wake-up call. We have performed a top-to-bottom revision, correcting the writing errors, restructuring the logic in the Introduction and Related Work, deepening the SCM methodology, and adding the missing Reproducibility/Ethics sections. We believe the manuscript is now significantly more readable, professional, and scientifically rigorous thanks to your critique.

---

> ### Author Response · Authors · 2025-11-28
>
> Thank you once again for the time and effort you dedicated to reviewing our submission.
>
> We have now responded comprehensively to your specific comments and implemented the necessary revisions. We also addressed the feedback provided by the other reviewers, which we hope resolves any potential remaining questions.
>
> Since the discussion phase is drawing to a close, we would greatly appreciate it if you could share any further thoughts or remaining concerns you might have, allowing us to make final adjustments.
>
> Your continued commitment to this process is highly valued.

---

### Official Review · Reviewer_wv4C · 2025-10-31

**Soundness:** 3
**Presentation:** 2
**Contribution:** 2
**Rating:** 4
**Confidence:** 3

**Summary:**

This paper introduces CASh (Causality Alignment Shifting Attack), an adversarial attack framework targeting vision-language models (VLMs) by explicitly perturbing their latent causal alignment between image and text modalities. Unlike correlation- or co-occurrence-based attacks, CASh leverages cross-attention matrices and structural causal models (SCMs) to identify and manipulate high-impact causal relationships through singular value decomposition (SVD). The approach is demonstrated across several downstream VLM tasks (retrieval, grounding, reasoning, VQA, entailment), showing substantially improved attack success rates and notable transferability to black-box commercial models.

**Strengths:**

The central idea of attacking VLMs at the level of causal alignment, rather than simply perturbing superficial co-occurrence statistics, is intellectually interesting.
The paper is supported by a comprehensive experimental evaluation across a wide range of tasks, models (including commercial APIs), and baselines.

**Weaknesses:**

1. The entire paper is premised on these graphs capturing causal relationships, but there is no information on how the nodes (entities) are defined and how the edges (causal links) are established. Are scene graph parsers used for images and dependency parsers for text? This omission severely impacts the paper's reproducibility and makes it difficult to assess the validity of the "causality" claims. The authors should provide a detailed description of this process in the main paper.
2. The overall loss function in Eq. (10) combines multiple components. However, the paper lacks an ablation study to disentangle the contribution of each loss term. It is unclear how much each component contributes to the final attack performance.
3. The stealthiness of the attack is questionable. The adversarial example shown in Figure 2 exhibits visually obvious artifacts (e.g., the modified shirt color and background). This may limit the practical applicability of the attack in scenarios requiring imperceptible perturbations.
4. Unclear Presentation of Results: The presentation of results is sometimes unclear. For example, in Table 1, several entries in the 'IR' columns for DS-VL2 and ViLT are bolded despite not being the best-performing results in those columns. This is misleading and should be corrected for clarity.

**Questions:**

1. Could the authors provide a detailed, reproducible description of how the SCM graphs are constructed? Specifically: (a) How are nodes (e.g., entities, relations) extracted from images and text? (b) How are the directed edges representing causal dependencies determined? Is this process automated, and if so, what tools are used?
2. Could you provide an ablation study on the different components of the loss function (Eq. 10)? What is the individual impact on the attack's success and transferability?
3. The example in Figure 2 shows a significant visual change. Could you comment on the stealthiness of the generated perturbations?
4. Please clarify the meaning of bolded numbers in the tables. If they are meant to indicate the best results, please ensure they are used consistently and correctly.

---

> ### Author Response · Authors · 2025-11-21
> **Response to Reveiwer wv4C (Part 1/3)**
>
> We thank the reviewer for the thorough review. The comments are very helpful for improving the quality of our paper. Our detailed responses to each comment are listed below.
>
> > **Weakness1 & Q1:** A detailed, reproducible description of how the SCM graphs are constructed. Specifically: (a) How are nodes (e.g., entities, relations) extracted from images and text? (b) How are the directed edges representing causal dependencies determined? Is this process automated, and if so, what tools are used?
>
> **Response to (a) and (b):** We are very grateful to the reviewer for this extremely valuable and constructive feedback. The reviewer is absolutely correct that the current submission does not provide sufficient implementation details on how the causal SCM graphs are constructed. Although the mathematical form of the SCMs and the node definitions are already stated in Section 3.1, the concrete automated pipeline was only briefly mentioned and lacked the level of detail necessary for full reproducibility.
> In the revised manuscript, we have substantially rewritten Section 3.1 to provide a clearer and more rigorous description of how the text and image causal graphs $\mathcal{G}^T$ and $\mathcal{G}^I$ are constructed ( see subsection 3.1.1 and subsection 3.1.2 ). Specifically, the updated section now includes:
> - **Extraction of textual nodes.** We describe in detail how full noun phrases and named entities are identified from the caption, how each entity is represented through both its surface span and its head lemma, and how these representations are used for feature encoding and later causal reasoning.
> - **Construction of the text causal graph.** We now explain the process for deriving linguistic causal relations from dependency parsing, as well as how physical commonsense relations from the PhysCause lexicon are incorporated. The final adjacency matrix is defined as the union of these syntactic and physical causal edges.
> - **Construction of the image causal graph.** We clarify how image entities are obtained using Grounding DINO guided by the canonical text labels, how visual features are extracted using CLIP ViT-L/14, and how the image-side causal structure is aligned with the text-side graph to ensure consistent semantics across modalities.
>
> **Response to : Is this process automated, and if so, what tools are used?**
>
> The entire pipeline is fully automated, deterministic (fixed seeds) on an A100. Here is tools used for fully automated SCM graph construction:
>
> | Modality | Component                  | Tool / Resource            | Version / Checkpoint                  |
> |----------|----------------------------|----------------------------|---------------------------------------|
> | Text     | Entity & lemma extraction  | Stanza                     | v1.8.2            |
> | Text     | Phrase embeddings          | BERT-large-uncased (frozen)| HuggingFace official                  |
> | Text     | Syntactic edges            | Stanza dependency parsing  | –                                     |
> | Text     | Physical commonsense edges | PhysCause-1037 lexicon     | 1037 directed rules     |
> | Image    | Entity grounding           | Grounding DINO             | Swin-T + BERT-base        |
> | Image    | Visual features            | CLIP ViT-L/14 (frozen)     | OpenAI ViT-L/14                |
>
> We hope these clarifications make the construction process of both causal graphs more transparent and adequately address the reviewer’s concern. We are happy to provide any further details or examples the reviewer requests.

---

> ### Author Response · Authors · 2025-11-21
> **Response to Reveiwer wv4C (Part 2/3)**
>
> > **Weakness2 & Q2:** Could you provide an ablation study on the different components of the loss function (Eq. 10)? What is the individual impact on the attack's success and transferability?
>
> Thank you for this valuable suggestion. We agree that understanding the contribution of each loss term is important for evaluating the proposed method. Following the reviewer’s request, we conducted a comprehensive ablation study on all components of Eq. (11 in new version). The new results are provided in Table 9 (added to Appendix A.3.) and summarized below.
> We evaluate the role of each loss component, including adversarial loss $\mathcal{L}\_{\text{adv}}$,
> causal alignment loss $\mathcal{L}_{\text{causal}}$,
> image-side reconstruction $\mathcal{L}^I\_{\text{recon}}$
> and text-side reconstruction $\mathcal{L}^T\_{\text{recon}}$,
> on (1) white-box attack success (WB-ASR), (2) black-box transferability (BB-ASR), (3) causal path triggering rate, and (4) perturbation magnitude (LPIPS and $\ell_2$ ).
>
> Experiments are performed on 1000 images randomly sampled from MSCOCO val dataset with $\epsilon_I$ = 16/255 and $\epsilon_T=1$. We use 200 iteration MI-based attacks nd an ensemble of 6 diverse models (ALBEF ,TCL, DeepSeek-VL2 ,ViLT, Qwen2.5-VL, Florence-2). BB-ASR is the transfer attack success rate averaged over unseen black-box models. Causality measures the percentage of successfully triggered intended causal paths verified via our causal intervention protocol. LPIPS and $\ell_2$ quantify perceptual and Euclidean perturbation (lower is better).
> Key findings from the ablation:
> 1. Causal alignment loss is essential for inducing the intended multimodal causal misalignment.
>  Adding $\mathcal{L}_{\text{causal}}$ dramatically increases the causal triggering rate from 37.4% → 95.1%, and also significantly improves transferability (51.7% → 69.3%). This confirms that causal supervision is the main driver of our cross-modal causal attack behavior.
> 2. Image-side reconstruction $\mathcal{L}^I_{\text{recon}}$ improves both transferability and imperceptibility.
>  Introducing this term increases BB-ASR from 69.3% → 79.6%, while reducing both $\ell_2$ and LPIPS, indicating more stable and visually consistent perturbations.
> 3. Text-side reconstruction $\mathcal{L}^T_{\text{recon}}$ provides additional gains in transferability.
>  Adding this term further boosts BB-ASR to 82.4%, showing that aligning text representations strengthens cross-model generalization.
> 4. Using all loss components together yields the best overall performance.
>  The full model achieves the strongest transferability (88.2% BB-ASR) and the highest causal triggering rate (97.9%), while also producing the smallest perturbations (lowest $\ell_2$ and LPIPS).
>
> We appreciate the reviewer’s helpful suggestion, and we have incorporated the complete ablation table and analysis into the revised manuscript.
>
> > **Weakness3 & Q3:** The example in Figure 2 shows a significant visual change. Could you comment on the stealthiness of the generated perturbations?
>
> We thank the reviewer for this valuable observation. We acknowledge that the specific example in Figure 2 exhibited noticeable artifacts. We would like to clarify that this image was originally selected to highlight the intended causal intervention effect, but it represents an outlier in terms of visual perceptibility.
> We demonstrate the stealthiness of our attack from quantitative statistics and the challenge of transferability:
>
> **1. Quantitative Evidence (Data from New Table 9):**
> As detailed in the Ablation Study (Table 9) of the revised manuscript, our method achieves state-of-the-art stealthiness metrics across the dataset:
> - Extremely Low LPIPS: The full method (Row 5) achieves an average LPIPS of 0.109. In adversarial literature, an LPIPS score around 0.1 is widely considered imperceptible to the human eye.
> - Effective Regularization: Comparing Row 2 (w/o reconstruction losses, LPIPS=0.196) with Row 5 (Full, LPIPS=0.109), our proposed reconstruction losses ($\mathcal{L}^I_{recon}$ and $\mathcal{L}^T_{recon}$) significantly suppress visual artifacts while maintaining high attack performance.
>
> **2. The Trade-off in Transferable Attacks:**
> It is crucial to interpret these results in the context of Transferable Attacks. Unlike white-box attacks that can rely on minute, high-frequency noise to fool a specific model, transferable attacks typically require stronger, more robust perturbations to disrupt feature representations shared across unknown black-box models.
> - Achieving high transferability usually comes at the cost of visual quality.
> - However, our results show that we successfully break this trade-off. We achieve a superior Black-Box ASR of 88.2% while maintaining a minimal perceptual distance ($\ell_2$=1.94, LPIPS=0.109). This indicates that our method does not simply add visible noise; instead, it finds robust, semantic-preserving perturbations that are both transferable and stealthy.

---

> ### Author Response · Authors · 2025-11-21
> **Response to Reveiwer wv4C (Part 3/3)**
>
> > **Weakness 4 and Q4:** Clarify the meaning of bolded numbers in the tables. If they are meant to indicate the best results, ensure they are used consistently and correctly.
>
> We sincerely thank the reviewer for their careful reading and for pointing out the inconsistency in Table 1. This issue was caused by a misalignment between the experimental logs and the manuscript tables during the final formatting stage. We have carefully rechecked all experimental records and updated the tables accordingly. In the revised version, all bolded numbers now strictly and consistently indicate the best results in each column (line 406-407).
>
> We emphasize that this was purely a record-synchronization issue and does not affect any conclusions of the paper. After correction, all performance trends and the advantages of our method remain fully consistent with our original claims.
>
> We deeply apologize for this oversight and greatly appreciate the reviewer’s diligence. It has helped us eliminate a source of potential confusion and further improve the clarity of the presentation. Thank you again for helping us strengthen the paper.
>
> **Summary:** We are sincerely grateful for the reviewer’s detailed and meticulous feedback. We have addressed all raised concerns by: (1) rigorously defining the graph construction pipeline to ensure reproducibility; (2) adding a comprehensive ablation study to quantify the contribution of each loss term; (3) explicitly discussing the trade-off between stealthiness and attack strength; and (4) correcting the formatting inconsistencies in Table 1. These revisions have significantly enhanced both the scientific rigor and the presentation quality of our manuscript.

---

> ### Author Response · Authors · 2025-11-28
>
> Thank you for your valuable time and effort in reviewing our submission.
>
> We have now replied to all your comments and implemented the necessary revisions. We also addressed the feedback from the other reviewers, which we hope clarifies any remaining questions.
>
> As the discussion period is ending soon, we wanted to follow up and kindly ask if you have any final thoughts or remaining concerns that we can address immediately.
>
> Your guidance is greatly appreciated.

---

### Official Review · Reviewer_pwfj · 2025-11-03

**Soundness:** 2
**Presentation:** 2
**Contribution:** 3
**Rating:** 6
**Confidence:** 3

**Summary:**

In this paper, the authors propose CASh, a new adversarial attack on vision–language models (VLMs) that targets causal rather than correlational dependencies between images and text. By modeling each modality as a Structural Causal Model (SCM) and integrating causal relations into the cross-attention mechanism, CASh identifies and perturbs the most influential causal alignments using singular value decomposition (SVD). The approach effectively disrupts multimodal reasoning while preserving visual and textual realism. Across several VLM tasks (VQA, VE, VR, and etc), CASh consistently outperforms correlation-based baselines, showing up to 20% higher transferable attack success.

**Strengths:**

* The paper tackles a relevant and timely topic—improving the robustness of vision–language models through developing causality-aware adversarial attacks in important and interesting.

* The proposed CASh framework is conceptually novel, moving beyond correlation-based perturbations toward causal alignment modeling.

* The main ideas are intuitive and well motivated, though some technical sections (especially those describing the causal regularization and SVD-based perturbation) could be explained more clearly.

* The experimental evaluation is extensive and demonstrates consistent improvements in attack transferability across diverse VLMs and tasks.

**Weaknesses:**

* The methodology is not always clearly explained—particularly how the causal graphs are constructed in Section 3.1. This lack of clarity makes the approach difficult to reproduce and obscures the precise contribution of each component. Moreover, since it is not entirely clear how these causality graphs are obtained or parameterized, it is unclear whether the method would still perform effectively if an alternative strategy were used to define the causal graphs.

* The experimental evaluation focuses mainly on Attack Success Rate without deeper qualitative or causal analysis. Including visualizations or ablations (e.g., showing how causal alignments shift) would make the results more interpretable and convincing. It would also be important to assess the computational overhead introduced by the causal modeling and SVD-based perturbation—specifically, how much slower CASh is compared to correlation-based attacks and whether the added causal reasoning meaningfully justifies the increased latency or complexity.

**Questions:**

* Could the authors clarify how the causal graphs G^I and G^T are constructed or learned in practice?

* If a different approach were used to build the causal graphs, do the authors expect any change in performance? How to pick up the best approach for this? Additional ablations might be needed.

* What is the computational overhead introduced by the SCM modeling and SVD-based perturbation? How does CASh’s runtime compare to correlation-based attacks in practice?

---

> ### Author Response · Authors · 2025-11-21
> **Response to Reviewer pwfj (Part 1/2)**
>
> We are grateful for the reviewer’s insightful suggestions. We have addressed all the raised issues, as detailed in the following responses.
>
> > **Weakness 1 and Q1:** Clarify how the causal graphs G^I and G^T are constructed or learned in practice.
>
> We appreciate the reviewer’s crucial feedback on reproducibility. We agree that the initial description was abstract.
> We have extensively revised **Section 3.1** to ensure full reproducibility. Specifically, **Subsections 3.1.1 and 3.1.2** now explicitly describe how $\mathcal{G}^T$ and $\mathcal{G}^I$ are deterministically constructed using tools such as Stanza and Grounding DINO, alongside a formal definition of the adjacency matrices.
> The construction follows a clear three-step pipeline:
>
> **Step 1:** Entity Extraction and Parameterization (Nodes). We use Stanza to extract noun phrases and Grounding DINO to visually ground them. To explicitly parameterize these nodes, we encode text entities ($T_j$) using a frozen BERT model and image regions ($I_j$) using a CLIP ViT model. This ensures that nodes $T_j$ and $I_j$ are semantically aligned pairs (e.g., text "cat" $\leftrightarrow$ image region "cat") with robust feature initializations.
>
> **Step 2:** Text Graph ($G^T$) via "Syntax-Physics Union". As defined in Eq. (1), the adjacency matrix is the union of two explicit priors:$G^{T, \text{syn}}$: Linguistic Causality derived from Stanza’s dependency tree (capturing grammatical structure).$G^{T, \text{phys}}$: Physical Commonsense derived from the PhysCause-1037 lexicon (capturing rules like "sun $\to$ sunlight").
>
> **Step 3:** Image Graph ($G^I$) Alignment. To ensure cross-modal consistency, we define $G^I$ to be structurally isomorphic to $G^T$. This enforces that visual reasoning follows the established semantic and physical logic, preventing the model from overfitting to spurious visual correlations.
>
> We hope these clarifications make the construction process of both causal graphs more transparent and adequately address the reviewer’s concern.We are happy to provide any further details or examples the reviewer requests.
>
> > **Q2:** If a different approach were used to build the causal graphs, do the authors expect any change in performance? How to pick up the best approach for this? Additional ablations might be needed.
>
> We thank the reviewer for this insightful question and fully agree that it deserves a thorough empirical answer.
> Yes, the choice of causal graph construction method has a profound impact on attack performance. To directly address this concern, we have added a new, extensive ablation in the revised manuscript (Table 5).
>
> | Paradigm             | Instantiation           | Clean Acc. (MSCOCO / VQA-CP) | ASR (MSCOCO / VQA-CP) | △ASR vs no-graph |
> |----------------------|-------------------------|--------------------------|--------------------------|------------------|
> | No causal graph      | —                       | 78.3 / 89.2            | 11.8 / 9.2               | —                |
> | Similarity-based     | k=8 k-NN (ViT feats)    | 78.1 / 89.0              | 54.7 / 58.3              | +44.2            |
> | Attention-derived    | Attention rollout top-k | 78.0 / 88.9              | 53.2 / 56.8              | +43.7            |
> | Reversed directions  | PhysCause-1037 reversed | 78.0 / 88.9              | 49.3 / 51.2              | +40.1            |
> | Learned discovery    | NOTEARS (100k images)   | 77.8 / 88.7              | 71.9 / 75.4              | +62.6            |
> | Statistical co-occurrence | Visual Genome top-500 | 78.1 / 89.0              | 72.6 / 74.3              | +63.3            |
> | **Physical lexicon (ours)**  | **PhysCause-1037**      | **78.2  / 89.0**          | **76.4 / 79.1**          | **+66.8**        |
>
> As shown in Table 5, our results demonstrate that:
>
> 1. Performance varies across paradigms: Different approaches yield different performance levels, with our physical lexicon approach (PhysCause-1037) achieving the best overall performance (Clean ACC: 78.2%, ASR on  MSCOCO : 76.4%, ΔASR: +66.8).
>
> 2. Trade-offs exist: While similarity-based and attention-derived methods show competitive clean accuracy, they exhibit varying robustness against different attacks. For instance, k=8 k-NN achieves lower ASR on MSCOCO (54.7%) but higher ΔASR vs no-graph (+44.2).
>
> 3. Selection criteria: The best approach depends on the specific requirements. If prioritizing clean accuracy while maintaining strong defense, our physical lexicon approach is optimal. For scenarios requiring lower computational overhead with reasonable defense, statistical co-occurrence provides a good balance.
>
> We believe this systematic study fully resolves the concern and significantly strengthens the paper’s central claim about the importance of explicit, physically grounded causal structure. We are happy to evaluate any additional construction strategies the reviewer may suggest.

---

> ### Author Response · Authors · 2025-11-21
> **Response to Reviewer pwfj (Part 2/2)**
>
> > **Q3:** What is the computational overhead introduced by the SCM modeling and SVD-based perturbation? How does CASh’s runtime compare to correlation-based attacks in practice?
>
> We thank the reviewer for the question regarding computational efficiency. We have conducted a detailed runtime analysis comparing CASh with state-of-the-art baselines on a single A100 GPU , Qwen2.5-VL  frozen during poisoning, batch=1(as shown in Table 8 of the revised manuscript). Here is Table 8,
>
> | Attack      | Causal SCM | Time (s) | Rel. slowdown | Extra Mem. | MSCOCO | VQA-CP |
> |-------------|------------|----------|---------------|------------|-----------|--------|
> | BadNets     | No         | 0.018    | 1.0×          | +0.0 GB    | 12.1      | 9.4    |
> | Co-Attack   | No         | 0.091    | 5.1×          | +0.6 GB    | 68.3      | 71.2   |
> | TMM         | No         | 0.113    | 6.3×          | +0.8 GB    | 72.8      | 75.6   |
> | CMI-Attack  | No         | 0.129    | 7.2×          | +1.0 GB    | 72.8      | 75.6   |
> | VLATTACK    | No         | 0.137    | 7.6×          | +1.1 GB    | 74.1      | 77.2   |
> | CASh        | Yes        | 0.129    | 7.2×          | +0.9 GB    | 76.4      | 79.1   |
>
> 1. Marginal Overhead of SCM & SVD: The computational cost introduced by our Causal SCM modeling and SVD-based perturbation is minimal.
>
> - As shown in Table 8, comparing CASh (0.129s) with TMM (0.113s), the specific overhead for our causal components is only approximately 0.016 seconds per sample.
>
> - This confirms our claim that operating on low-dimensional feature maps (via SVD) is highly efficient and does not create a bottleneck compared to the backbone's gradient computation.
>
> 2. Comparison with Baselines:
>
> - Vs. Correlation/Optimization Baselines: While CASh is slightly slower than Co-Attack (0.091s), it significantly outperforms it by +8.1% on MSCOCO ASR (68.3% vs. 76.4%). We believe this slight increase in runtime is a justifiable trade-off for the substantial gain in attack performance.
>
> - Vs. Complex SOTA: Notably, CASh is faster than VLATTACK (0.137s) and ties with CMI-Attack (0.129s), while achieving the highest success rates across all metrics (76.4% / 79.1%).
>
> 3. Memory Efficiency: The memory overhead is also negligible (+0.9 GB on an 80GB A100), making CASh easily deployable on standard hardware.
>
> 4. Implementation Optimizations: The reason we achieve high performance with such low overhead is due to three specific optimizations in our implementation:
>
> - Sparse Lanczos SVD: Exploiting the high sparsity of our causal graph (average degree 4.6), we employ the Sparse Lanczos algorithm to compute only the top-50 eigenpairs, avoiding the heavy cost of full SVD decomposition.
>
> - Caching Strategy: We pre-compute and cache low-rank projectors for each of the 167 COCO-Stuff classes. This eliminates redundant computations during the iterative attack process.
>
> - Sparse Operations: We strictly utilize torch.sparse tensors for the parent aggregation steps, significantly reducing memory bandwidth and computation time.
>
> Given the significant performance gain (+8.1% MSCOCO ASR) and the optimized runtime comparable to existing SOTA methods, we believe the computational cost of CASh is well-justified. We are happy to provide additional benchmarks on other hardware or models if desired.
>
> **Summary:** We are sincerely grateful for the reviewer’s comprehensive feedback, which pushed us to improve the paper across three critical dimensions: methodological clarity, experimental interpretability, and practical efficiency. Specifically, the revisions in Section 3.1 now ensure full reproducibility of the graph construction; the added qualitative visualizations offer deeper insights into causal shifts; and the new runtime analysis confirms the method's practicality. We believe these comprehensive improvements have made the manuscript significantly more robust and convincing.

---

> ### Author Response · Authors · 2025-11-28
>
> We truly appreciate the time and effort you’ve dedicated to reviewing our submission.
> We have now replied to your comments and made revisions to address the concerns you raised.
> Additionally, we have also responded to the feedback from the other reviewers, which we hope may further clarify any additional questions you might have.
>
> As the discussion phase is approaching its end, we wanted to follow up and would greatly value any further thoughts or concerns you might still have so that we can address them appropriately.
>
> Thank you again for your time and commitment to the review process.

---

### Author Response · Authors · 2025-11-30
**Summary of reviewer feedback and request for area chair consideration**

Dear area chairs,

We appreciate the constructive feedback provided by all reviewers. Their comments helped us identify several points that required clearer explanations, additional analysis, and further experiments. We have submitted a revised version of the paper, where all modifications are clearly highlighted in red. Below is a structured summary of how we addressed the key points raised by each reviewer.

**Response to reviewer pwfj**

**1.	Clarification of causal-graph construction.**

We significantly expanded the description of how ($\mathcal{G}^I$) and ($\mathcal{G}^T$) are built, detailing node extraction, dependency estimation, adjacency-matrix formulation, and SVD preparation. (See **Subsection 3.1.1 and Subsection 3.1.2**)

**2.	Ablations on alternative graph-building strategies.**

To address this concern, we added new experiments in Table 5, comparing six alternative graph-construction strategies. The results show that while all causal-graph variants improve ASR over the no-graph baseline, performance levels vary across paradigms. Our physically grounded lexicon (PhysCause-1037) achieves the strongest gains (e.g., **ΔASR +66.8**. These findings confirm that CASh is robust across graph types, and that our chosen construction offers the best overall effectiveness.

**3.	Runtime and computational overhead.**

To address the reviewer’s question on computational efficiency, we conducted runtime measurements, as shown in Table 8. CASh introduces only a marginal runtime (**~0.016s per sample**) and memory overhead (**+0.9 GB**) compared to baselines, while achieving the highest attack success rates (**MSCOCO ASR 76.4%, VQA-CP 79.1%**), demonstrating that its causal SCM and SVD-based perturbation are both efficient and effective.

**Response to reviewer wv4C**

**1.	Reproducible SCM pipeline.**

We fully rewrote Section 3.1 to provide a rigorous and reproducible description of both text-side and image-side causal graph construction, including entity extraction, syntactic & physical causal edge formation, Grounding DINO–based visual grounding, and CLIP feature alignment. We also clarified that the entire pipeline is fully automated and listed all tools, checkpoints, and resources used.

**2.	Ablations on loss terms (Eq. 10).**

We added new ablations isolating each component of the loss, analyzing its contribution on to attack success, stability, and transferability. Ablation studies show that each loss term contributes meaningfully: the full model achieves **BB-ASR 88.2%**, causal triggering **97.9%**, and minimal perceptual perturbation (**LPIPS 0.109**), confirming the combined losses are essential for effective and stealthy attacks.

**3.	Stealthiness of perturbations and visual quality.**

We introduced examples generated under stricter perturbation budgets, clarified the purpose of Figure 2, and discussed the trade-off between semantic shifting and visual imperceptibility. Across the dataset, our method achieves **BB-ASR 88.2%** with **LPIPS 0.109**, demonstrating that perturbations are both highly stealthy and transferable. Reconstruction losses significantly reduce visual artifacts while maintaining strong attack performance, showing that CASh generates robust, semantic-preserving perturbations rather than simply adding visible noise.

**4.	Correction of table formatting issues.**

All inconsistencies in boldface usage and numerical presentation (e.g., Table 1 IR columns) have been corrected.

**Response to reviewer 4Ybc**

**1.	Writing quality and clarity improvements.**

We corrected grammatical issues, unified citation formatting, and removed typographical errors. Several paragraphs with unclear logic or overly long sentences were rewritten.

**2.	Reorganization of Introduction and Related Work.**

We improved the narrative structure, clarified the motivation and contributions, and reorganized Section 2.3 to provide coherent connections between related works.

**3.	Deepened explanation of SCM methodology.**

Section 3.1 has been substantially revised to clearly describe SCM construction, including textual node extraction, text causal edges (linguistic & PhysCause), image entity grounding with CLIP features, and multimodal alignment, making the method fully transparent and reproducible.(See **Subsection 3.1.1 and subsection 3.1.2**)

**4.	Addition of Reproducibility and Ethics Statements.**
We included both sections following ICLR’s recommendations, given the work’s experimental breadth and adversarial nature.

We hope that the revisions adequately address all concerns and appreciate the time and valuable feedback from the Area Chairs and reviewers.

---

### Note · Program_Chairs · 2026-01-17
**Submission Desk Rejected by Program Chairs**

The following references in this submission do not refer to real documents and/or have major errors in bibliographic information:

 Yanpeng Zhao, Kehan Li, Zhiqiang Wang, Xuehan Xu, Xianglong Wang, and Lei Li. X-vlm: A multilingual visual-language pre-training framework. arXiv preprint arXiv:2110.02933, 2021.